

# Impact of crop field burning and mountains on heavy haze in the North China Plain: A case study

X. Long[1,2], X. X. Tie[1,3,4,*], J. J. Cao[1,5], R. J. Huang[1,6,*], T. Feng[1], N. Li[1,7], S. Y. Zhao[1], J. Tian[1], G. H. Li[1], Q. Zhang[8]

5   [1]Key Lab of Aerosol Chemistry & Physics, SKLLQG, Institute of Earth Environment, Chinese Academy of Sciences, Xi'an, 710061, China

[2]University of Chinese Academy of Sciences, Beijing, 100049, China

[3]CAS Center for Excellence in Urban Atmospheric Environment, Xiamen, China

[4]National Center for Atmospheric Research, Boulder, CO, 80303, USA

10   [5]Institute of Global Environmental Change, Xi'an Jiaotong University, Xi'an, 710049, China

[6]Laboratory of Atmospheric Chemistry, Paul Scherrer Institute (PSI), 5232 Villigen, Switzerland.

[7]Department of Atmospheric Science, National Taiwan University, Taipei, 10617, Taiwan

[8]Center for Earth System Science, Tsinghua University, Beijing, 100084, China

*Correspondence to*: X. X. Tie (xxtie@urcar.edu) R. J. Huang (rujin.huang@ieecas.cn)





**Abstract.** Crop field burning (CFB) has important effects on air pollution in China, but it is seldom quantified and reported in a regional scale, which is of great importance for the control strategies of CFB in China, especially in the North China Plain (NCP). With the provincial statistical data and open crop fires captured by satellite (MODIS), we extracted a detailed emission inventory of CFB during a heavy haze event from $6^{th}$ to $12^{th}$ October 2014. A regional dynamical and chemical model (WRF-Chem) was applied to investigate the impact of CFB on air pollution in NCP. The model simulations were compared with the in situ measurements of $PM_{2.5}$ (particular matter with radius less than 2.5 μm) concentrations. The model evaluation shows that the correlation coefficients (R) between measured and calculated values exceeds 0.80 and absolute normalized mean bias (NMB) is no more than 14%. In addition, the simulated meteorological parameters such as winds and planetary boundary layer height (PBLH) are also in good agreement with observations. The model was intensive used to study (1) the impacts of CFB and (2) the effect of mountains on regional air quality. The results show that the CFB occurred in southern NCP (SNCP) had significant effect on $PM_{2.5}$ concentrations locally, causing a maximum of 35% $PM_{2.5}$ increase in SNCP. Because of south wind condition, the CFB pollution plume is subjective a long transport to northern NCP (NNCP-with several mega cities, including Beijing of the capital city in China), where there are no significant CFB occurrences, causing a maximum of 32% $PM_{2.5}$ increase in NNCP. As a result, the heavy haze in Beijing is enhanced by the CFB occurred in SNCP. Further more, there are two major mountains located in the western and northern NCP. Under the south wind condition, these mountains play important roles in enhancing the $PM_{2.5}$ pollution in NNCP through the blocking and guiding effects. This study suggests that the $PM_{2.5}$ emissions in SNCP region should be significantly limited in order to reduce the occurrences of heavy haze events in NNCP region, including the Beijing City.

**Key words:** crop field burning; mountain affect; $PM_{2.5}$ pollution; WRF-Chem; North China Plain (NCP)





## 1 Introduction

Biomass burning processes contribute large amounts of particulate matter to air pollution (He et al., 2015b;van der Werf et al., 2006). Crop field burning (CFB) is important for biomass burning (Yevich and Logan, 2003), especially in agricultural countries such as China, the CFB accounts for a high

proportion of open fires and represents a severe threat to air quality (Cao et al., 2008). Indeed, CFB have already been banned, but the local enforcement of regulation is limited (Zhang and Cao, 2015). Large amounts of crop residues are still burned during the post-harvest seasons (Yan et al., 2006;Streets et al., 2003), and extensive crop fires are concentrated in the North China Plain (NCP) (Huang et al., 2012), where have been frequently suffering haze events in recent years (Yang et al.,

2015;Jiang et al., 2015;Wang et al., 2013;Wang et al., 2012).

Previous studies have reported the importance of CFB contribution to $PM_{2.5}$ of the Pearl River delta (PRD) (Wang et al., 2007;Zhang et al., 2010;He et al., 2011), the Yangtze River delta (YRD) (Cheng et al., 2014) and the NCP region (Wang et al., 2007;Li et al., 2010;Cheng et al., 2013;Yang et al., 2015). The impact of CFB is regional, and inter-province transported air pollutants significantly affects

regional $PM_{2.5}$ levels and air quality (Cheng et al., 2014). A recent study reports that CFB and regional transport partly illustrates the key process of haze formation in October 2014, especially on Oct. 6[th] (Yang et al., 2015), but it is lack of study for the quantitative effect. However, related quantification studies are of great importance for the control strategies of CFB in China.

In this study, we analyzed a heavy haze episode occurred in NCP region from "LT" 12:00 6[th] to 00:00

12[th] October in 2014, during which CFB were captured by Moderate Resolution Imaging Spectroradiometer (MODIS). Meanwhile, the location and topographic feature of NCP provide a good opportunity to study the impact of mountains on the air pollution. We aims to: (1) analyze the characteristics of the air pollution based on $PM_{2.5}$ concentration; (2) extract a more detailed CFB emission inventory with higher temporal/spatial resolution based on the provincial statistical data and

MODIS observations; (3) quantify the contributions of CFB on the evolution of $PM_{2.5}$ concentration and (4) study the effect of mountains (especially the Taihang Mountains and Yanshan Mountains) on the pollution transport during the haze episodes.





## 2 Description of data

### 2.1 Geographical Location

In order to study the effect of CFB on local and regional air pollution, the research domain locates in eastern China, covering a large regional area (more than 10 provinces) (see **Fig. 1a**). The NCP region is

in the middle of the research domain, with two mountains in the north and west. The Yanshan Mountains locate in the north of NCP with east-west directions, and the Taihang Mountains locate in the west of NCP with southwest-northeast directions (**Fig. 1b**). **Figure 1c** displays the distribution of online sampling sites and CFB captured by MODIS during the haze episodes. According to crop fires, topographic conditions, industrial and agricultural developments, we defined two regions. One is the

north part of NCP (NNCP), including two mega cities (Beijing and Tianjin), and the north part of Hebei province, where only few CFB occurred. Another is the south part of the NCP (SNCP), where substantial crop fires occurred during the haze episodes (as shown in **Fig. 1c**). Because of the severe haze problem in the capital city of China (Beijing), one of the main focuses is to study the long-range transport of CFB from SNCP to NNCP.

### 2.2 PM$_{2.5}$ Measurements

The hourly PM$_{2.5}$ mass concentration were continually monitored by the Ministry of Environmental Protection (MEP) of China (http://datacenter.mep.gov.cn), including 5 sites in NNCP and 7 sites in SNCP (indicated by green crosses in **Fig. 1c**). The data was updated from the website: http://www.pm25.in/. **Table 1** summarizes the site information and the measured PM$_{2.5}$ concentrations.

During the study period, the averaged PM$_{2.5}$ concentrations are 200.0 μg m$^{-3}$ and 184.1 μg m$^{-3}$ in NNCP and SNCP, respectively. The measured PM$_{2.5}$ concentrations are much higher than class II standard (daily mean of 75 μg m$^{-3}$), indicating an occurrence of heavy pollution event. It is worth to note that the highest PM$_{2.5}$ concentrations occurred along the foothill of the Taihang Mountains. For example, at the sites of BJ, BD, SJZ and XT, PM$_{2.5}$ concentrations are 245.5, 287.7, 257.9, and 320.1 μg m$^{-3}$,

respectively. The mean PM$_{2.5}$ concentration in these 4 sites is 277.8 μg m$^{-3}$, much higher than 147.2 μg m$^{-3}$ averaged from the other sites.

### 2.3 Meteorological conditions

The reanalysis meteorological data, including wind direction, wind speed and planetary boundary layer





height (PBLH) were obtained from the European Centre for Medium-range Weather Forecasts (ECMWF), with a spatial resolution of 0.125° × 0.125°. The data is available at: http://www.ecmwf.int/products/data/. The averaged wind directions and wind speed are displayed in **Table 1**. It shows that during the haze episode, the mean wind directions are 174.8° in NNCP 165.2° in

SNCP, and the average wind speeds are 2.4 m s$^{-1}$ in both NNCP and SNCP. The meteorological data suggests that the prevailing winds are continually southerly winds, with weak wind speeds, which are in favor to form haze events. The south winds led to pollution transport from SNCP to NNCP, and generally produced high air pollutions in the Beijing City (Tie et al., 2015).

### 3 Methods

**3.1 Model description**

The Weather Research and Forecasting Chemical model (WRF-Chem) was used to simulate the spatial and temporal variability of PM$_{2.5}$ concentration. The WRF-Chem model is a state-of-the-art regional dynamical/chemical transport model with detailed description available at https://www2.acom.ucar.edu/wrf-chem. The model configuration includes simultaneous calculation of

dynamical parameters (winds, temperature, boundary layer, clouds, etc.), transport (advective, convective, and diffusive), dry deposition (Wesely, 1989), wet deposition, gas phase chemistry, radiation and photolysis (Tie et al., 2003;Madronich and Flocke, 1999), and online calculation of biogenic emission (Guenther et al., 1994). The gas-phase chemistry was represented in the model by the modified RADM2 (Regional Acid Deposition Model, version 2) gas-phase chemical mechanism

(Stockwell et al., 1990;Chang et al., 1987). In the present study, we used the CMAQ (version 4.6) aerosol module developed by US EPA (Binkowski and Roselle, 2003). We also used the Yonsei University (YSU) PBL scheme, which utilizes counter-gradient terms to represent fluxes and explicitly considers the entrainment effect to calculate the PBL heights (Hong et al., 2006). Meanwhile, the model employed the Lin microphysics scheme (Lin et al., 1983), the Noah land-surface model (Chen

and Dudhia, 2001), the long-wave radiation parameterization (Mlawer et al., 1997), and the shortwave radiation parameterization (Dudhia, 1989). The model has been successfully applied in several regional pollution studies in the globe (Tie et al., 2009;Tie et al., 2007;He et al., 2015a).

The model resolution is 6 × 6 km in 1200 × 1800 km domain centered in (117°E, 39°N). Vertical layers





extended from the surface to 50 hPa, with 28 vertical layers, involving 7 layers in the bottom of 1 km.

The meteorological initial and boundary conditions were gathered from NCEP FNL Operational Global

Analysis data. The lateral chemical initial conditions were constrained by a global chemical transport

model-MOZART4 (Model for Ozone and Related chemical Tracers, Version 4) 6-hour output

5    (Emmons et al., 2010;Tie et al., 2005). For the episode simulations, the spin-up time of the WRF-Chem

model is 12 hours.

The surface emission inventory used in this study was obtained from the Multi-resolution Emission

Inventory for China (MEIC) (Zhang et al., 2009), which is an update and improvement for the year

2010 (http://www.meicmodel.org). The emission inventory estimated only anthropogenic emission

such as non-residential sources (transportation, agriculture, industry and power) and residential sources

related to fuel combustions, we added emission from CFB in the present study.

**3.2 Crop field burning emission**

To estimate the CFB, we analyzed the annual and monthly number of crop fire events captured by

MODIS in the research domain from 2006 to 2014. In the NCP region, the CFB is gradually increasing

since 2008, from the minimum fire events of 12, 000 times in 2008 to 27, 000 times in 2014 (**Fig. 2a**),

suggesting that the CFB is not efficiently controlled in the region. The burning events mostly occurred

in June and October due to the post-harvest activities (**Fig. 2b**). The strong seasonal variation suggests

that the emission from CFB is very important, but only occurred in particular months (June and Oct.) to

the pollution events in NCP. In order to have the detailed horizontal distribution of the pollutant

emissions from CFB, we elaborated a method to generate emission inventory using the satellite data of

"MODIS Thermal Anomalies/Fire product (MOD/MYD14DL)". The MOD/MYD14DL product

detected small opening fires (<100 $m^2$), with daily temporal resolution (Giglio et al., 2003), and located

fire activities (van der Werf et al., 2006).

We estimated the CO emission of CFB using the annual provincial statistical data (Streets et al.,

2003;Cao et al., 2008;Zhang et al., 2008;Ni et al., 2015). The provincial emission of crop residues

burning can be calculated by Eq. (1):

$$E_{i,CO} = P_i \times R \times F_i \times CE \times EF_{co} , \qquad (1)$$

where $E_{i,CO}$ stands for CO emission from CFB of *i-th* province; $P_i$ is provincial crop production; $R$

is crop-specific-residue-to-production ratio (dry matter); $F_i$ is provincial crop-specific percentage of





crop residues burned in the field; $CE$ is percentage of combustion efficiency; $EF_{co}$ is the emission

factors of CFB.

Furthermore, the CO emission was temporally and spatially allocated according to the CFB activities

(Huang et al., 2012), which was defined as MOD/MYD14DL active fires occurred over the cropland

classification of the MODIS Combined Land Cover Type product (Friedl et al., 2010). The detailed CO

emission of *k-th* grid ($E_k$) was calculated using Eq. (2):

$$E_{k,CO} = \frac{FC_k}{FC_i} \times E_{i,CO} \,, \qquad (2)$$

where $FC_k$ is the total fire counts in *k-th* grid, and $FC_i$ is the total fire counts in *i-th* province.

Based on the spatial and temporal emission of CO, the emissions of various gaseous and particulate

species ($E_{spec1}$) were calculated by the Eq. (3) and individual chemical compounds ($E_{spec2}$) were

calculated by Eq. (4).

$$E_{k,spec1} = \frac{EF_{spec1}}{EF_{CO}} \times E_{k,CO} \,, \qquad (3)$$

$$E_{k,spec2} = E_{k,NMOC} \times \text{scale}, \qquad (4)$$

where $E_{k,spec1}$ and $E_{k,spec2}$ are the *k-th* grid emission of the specify WRF-Chem species; $EF_{spec1}$

and $EF_{CO}$ are the emission factors of CFB; $E_{k,NMOC}$ is the *k-th* grid emission of NMOC calculated by

Eq. (3); *scale* is the value to convert NMOC emissions to WRF-Chem chemical species. The emission

factors for gaseous and particulate species, and scales to convert NMOC emissions to WRF-Chem

chemical species from CFB were taken from available datasets (Wiedinmyer et al., 2011;Akagi et al.,

2011;Andreae and Merlet, 2001) (see **Table 2**).

**4 Results and discussions**

**4.1 Evaluate the Crop field burning emission**

The provincial CO emissions of CFB were estimated based on Eq. (1) (see **Supplementary Table S1)**.

In order to evaluate the estimate of CFB emissions, we compared our result to previous studies. In our

evaluation, the total CO emission of CFB in China is 8481 Gg in 2012. This result is comparable to

25 previous published results of Cao et al. (2008) for 8241 Gg in 2002 and Huang et al. (2012) for 4360

Gg in 2006.

In this case study, according to the crop fires detected by the MODIS in NCP during the haze episode,

a large amount of agriculture residues burning activities occurred in SNCP, including provinces of





Henan with 61% and Shandong with 22% (see **Fig. 3** and **Table 3**). The most burning occurred on the

Oct. 6[th] with 56%, and it decreased to 18% on Oct. 7[th] (**Table 3**). We obtained the daily CO emission of

CFB depending on Eq. (2). **Fig. 3** displays the CFB and related CO emission on Oct. 6[th] and 7[th] when

the most CFB occurred.

Emission of chemical species required by the WRF-Chem model were calculated using Eq. (3) and (4).

**Table 4** shows the gaseous and particulate species emissions from CFB on Oct. 6[th] and 7[th], including

the mega cities of Beijing and Tianjin, and provinces of Hebei, Henan and Shandong in NCP. Most of

the pollutants are emitted from Henan in SNCP, accounting for 73% on Oct. 6[th] and 65% on Oct. 7[th].

Large amounts of pollutions emitted from CFB on Oct. 6[th], producing more than 5.4 Gg $PM_{2.5}$ and

103.9 Gg CO (1 Gg = $10^9$ g).

**4.2 Statistical characteristics of the evaluation**

The characteristics of the haze pollution was defined by $PM_{2.5}$ concentration, which is significantly

affected by the local wind field and PBLH in the NCP region (Tie et al., 2015). In order to evaluate the

model performance, the model simulation was intensive compared with the measured results in both

$PM_{2.5}$ concentration and meteorological parameters (wind speed, wind direction, and the PBLH). The

normalized mean bias (NMB) and correlation coefficient (R) were used to quantify the performance.

$$NMB = \frac{\sum_{i=1}^{N}(P_i - O_i)}{\sum_{i=1}^{N} O_i}, \tag{5}$$

$$R = \frac{\sum_{i=1}^{N}(P_i - \bar{P})(O_i - \bar{O})}{[\sum_{i=1}^{N}(P_i - \bar{P})^2 \sum_{i=1}^{N}(O_i - \bar{O})^2]^{\frac{1}{2}}}, \tag{6}$$

where $P_i$ is the predicted results and $O_i$ represents the related observations. N is the total number of

the predictions used for comparisons. Meanwhile, $\bar{P}$ and $\bar{O}$ are the average prediction and related

mean observation, respectively.

**Figure 4** shows the measured and calculated temporal variations of regional averaged $PM_{2.5}$

concentration, wind speed, wind direction and PBLH. The WRF-Chem model reproduced the pollution

episode with a good agreement with observations. The correlation coefficients (R) of simulated and

measured $PM_{2.5}$ concentration are 0.87 in NNCP and 0.80 in SNCP. The NMB are -14% in NNCP and

-3% in SNCP. The relative high NMB of -14% is mainly due to the negative bias in S3, which may be

resulted from cloud contamination (**Supplementary Fig. S1**), and it has few impacts on the

contribution of the CFB since few open crop fires occurred during that time. The comparisons between





simulated and observed wind fields show good agreements (**Fig. 4b and 4c**), with all the R being higher than 0.65, and the absolute NMB being no more than 15%. In addition, the R of PBLH are larger than 0.88 and the NMB are smaller than 10% in both NNCP and SNCP (**Fig. 4d**).

### 4.3 Characteristics of the heavy pollution events

According to the evolution of $PM_{2.5}$ concentration (see **Fig. 4a**), the haze episode can be divided into three stages: (I) pollution formation stage (S1, 12:00 6th - 00:00 8th), (II) pollution outbreak stage (S2, 00:00 8th - 00:00 10th) and (III) pollution clear stage (S3, 00:00 10th - 00:00 12th). The major characteristics of each stage are briefly summarized below. The detailed observations are followed by related simulations in bracket.

- S1 (pollution formation): It is dominated by a strong southerly wind, with mean wind speed of 2.5 (2.7) m s$^{-1}$ in NNCP and 3.0 (3.6) m s$^{-1}$ in SNCP. The pollution is continuously transported from SNCP to NNCP, leading to pollutants accumulation in NNCP, which is characterized by the steady rising $PM_{2.5}$ concentration in NNCP from 20.6 (39.6) µg m$^{-3}$ (at 12:00 Oct. 6th) to 242.7 (218.7) µg m$^{-3}$ (at 00:00 Oct. 8th) (**Fig. 4 a1**).

- S2 (pollution outbreak): The S2 is a relative stable period of heavy pollution with averaged $PM_{2.5}$ concentration of 252.0 (241.4) µg m$^{-3}$ in NNCP and 214.1 (235.1) µg m$^{-3}$ in SNCP, which are much higher than those in other stages. It was related to relative lower wind speed and PBLH, which are 2.1 (2.2) m s$^{-1}$ and 785 (908) m in NNCP, and 2.5 (2.9) m s$^{-1}$ and 909 (921) m in SNCP.

   - S3 (pollution clear): During S3, the southerly gradually decrease, and turn to northerly at the end of

S3. The clean air from the north region of NNCP obviously improves air quality. Compared with S2, the averaged $PM_{2.5}$ concentrations are both decreased in NNCP and SNCP.

There were several important issues shown in the results, and should be addressed. (1) The $PM_{2.5}$ concentrations are extremely high during the S2 period, and the daily average concentrations are

exceed the Chinese National Standard (75 µg m$^{-3}$) by 2-3 times. (2) The pollutions are severe in a large region (occurred in both NNCP and SNCP). (3) During the S1 and S2 periods, there is a time lag between SNCP and NNCP for $PM_{2.5}$ concentrations. Because it is a south wind direction, it shows the important impact of long-range transport of $PM_{2.5}$ particles from the SNCP to NNCP.





### 4.4 Contributions of crop field burning

Model sensitive studies were conducted to separate the individual contribution of CFB on the heavy aerosol pollution. Two model simulations were performed, i.e., one with both anthropogenic and CFB emissions while the other with only anthropogenic emission. We calculated PM$_{2.5}$ distributions by including crop fire emissions (anthropologic and CFB) and excluding crop field emissions (only anthropologic). The contributions were quantified by regional averaged contribution in mass concentration ($CMPM_{2.5}$) and daily averaged contribution ratio ($\overline{RPM_{2.5}}$).

$$CMPM_{2.5} = TPM_{2.5} - APM_{2.5}, \tag{7}$$

$$\overline{RPM_{2.5}} = \frac{\overline{CMPM_{2.5}}}{\overline{TPM_{2.5}}}, \tag{8}$$

where $TPM_{2.5}$ represents the simulated PM$_{2.5}$ concentrations considering total emission; $APM_{2.5}$ denotes the simulated PM$_{2.5}$ concentrations only considering anthropologic emissions. $\overline{CMPM_{2.5}}$ and $\overline{TPM_{2.5}}$ are daily averaged value for $CMPM_{2.5}$ and $TPM_{2.5}$, respectively.

**Figure 5** displays the regional observed and simulated PM$_{2.5}$ concentrations considering total emissions (anthropologic and CFB) and only anthropologic emissions. It is clearly shown that the CFB had important contributions to PM$_{2.5}$ in both NNCP (**Fig. 5a)** and SNCP **Fig. 5b)**. This is also proved by the daily averaged contribution ratio ($\overline{RPM_{2.5}}$) of CFB (**Table 5**). The high values of $\overline{RPM_{2.5}}$ in SNCP occur on Oct 6[th] with 35% and on 7[th] with 17%, when a large amount of CFB happened. Simultaneously, the high values of $\overline{RPM_{2.5}}$ in NNCP occur on Oct 7[th] with 32% and 8[th] with 10%, showing a later occurrence (one day-lag) than that in SNCP. The one-day lag suggested that the plume with CFB could be transported from SNCP (where CFB occurred) to NNCP.

The detailed hourly contributions of CFB to PM$_{2.5}$ mass concentration ($CMPM_{2.5}$) are displayed in **Fig. 6**. The values of $CMPM_{2.5}$ in NNCP are generally lag synchronized with that in SNCP, such as P$_{N1}$ versus P$_{S1}$ and P$_{N2}$ versus to P$_{S2}$ (**Fig. 6a and 6b**). Apparently, the lagged time is not constant and varied with the wind field. The specific details performed relaxed lag synchronized, especially the P$_{N2}$ versus to P$_{S2}$. **Figure 6** further indicates that the CFB contribution in SNCP is mainly due to local emission, while contribution in NNCP is largely resulted from regional transport. Indeed, day-averaged transport contribution to PM$_{2.5}$ from CFB in NNCP can be as high as 32% (see **Table 5**). Such a high transported contribution indicates that the CFB has not only a local pollution, but also has significant



regional impact on air pollution.

Moreover, the $CMPM_{2.5}$ in SNCP drops much faster than that in NNCP (see **P2 in Fig. 6c**). To clearly show the time evolution of the effect of CFB on $PM_{2.5}$ concentration, four time-points were defined in **Fig. 6c**, such as T1 (23:00 6[th]), T2 (05:00 7[th]), T3 (20:00 7[th]) and T4 (19:00 8[th]). It shows that at T1, there is a large CFB (in P1), and the $CMPM_{2.5}$ is the highest (76.1 μg m$^{-3}$) in SNCP, but with a low value (6.2 μg m$^{-3}$) in NNCP. At T2, the $CMPM_{2.5}$ is decrease (53.7 μg m$^{-3}$) in SCNP, but has high value (44.3 μg m$^{-3}$) in NNCP (near the transition between P1 and P2). At T3, the $CMPM_{2.5}$ in SNCP rapidly decreased to a low value (25.6 μg m$^{-3}$), but the value is the highest (48.7μg m$^{-3}$) in NNCP. At T4, the $CMPM_{2.5}$ are low in both SCNP (8.7 μg m$^{-3}$) and NNCP (11.8 μg m$^{-3}$), indicating the effect of CFB largely decreases. The values of $CMPM_{2.5}$ in NNCP are higher than that in SCCP from T3 to T4, indicating the longer effect of CFB on $PM_{2.5}$ concentration in NNCP than in SNCP (in P2).

**Figure 7** shows the horizontal distributions of $TPM_{2.5}$ and $CMPM_{2.5}$ at T1, T2, T3 and T4, and the related regional statistical results of $CMPM_{2.5}$ is displayed in **Table 6**. It shows that at T1 the massive local pollutants are emitted from CFB in SNCP and it had not been significantly transported to NNCP. The values of $CMPM_{2.5}$ are high in SNCP with interquartile range of 23-109 μg m$^{-3}$ ([Q1-Q3]), whereas in NNCP, the values of $CMPM_{2.5}$ are low with interquartile range of 0-10 μg m$^{-3}$. At T2, high $CMPM_{2.5}$ values with interquartile range of 10-60 μg m$^{-3}$ remains in both SNCP and NNCP, suggesting that a large amount of CFB pollutants emitted from SNCP and had been transported to NNCP. At T3, values of $CMPM_{2.5}$ rapidly reduce in SNCP with interquartile range of 5-36 μg m$^{-3}$. It is worth to note that the high $CMPM_{2.5}$ values with interquartile range of 28-72 μg m$^{-3}$ are still remained in NNCP. The highest values of $TMPM_{2.5}$ are along the foothill of the Taihang Mountains (see **Left panels of Fig.7**), indicating the influence of mountains, and the detailed effects of mountains were analyzed in the following sections. At T4, the pollutants contributed by CFB largely decreases in both SNCP and NNCP. More details about the statistical results of $CMPM_{2.5}$ are shown in **Table 6**.

**4.5 Impact of mountains**

To clarify the impact of mountains on $PM_{2.5}$ pollution, sensitivity model experiments were conducted to quantify the impacts of the Taihang Mountains (referred as R-T), the Yanshan Mountains (R-Y) and both (R-TY) on the heavy pollution in NCP. We removed the mountains from the model calculation, in





which, the altitude of mountains were reduced to the averaged altitude of NCP (30 m). With the reduction of altitudes of the topography, the dynamical conditions calculated from WRF-Chem changed, which affect pollutions transport, especially along the foothill of mountains. The differences between the simulations with or without mountains showed the net effect of the topography on $PM_{2.5}$

concentration, which was calculated using Eq. (9). And the sensitive configuration and related enclosing scope are displayed in **Supplementary Fig. S2.**

$$IPM_{2.5} = RPM_{2.5} - TPM_{2.5}, \tag{9}$$

where $IPM_{2.5}$ is the net impacts of mountains on $PM_{2.5}$; $RPM_{2.5}$ denotes the simulated $PM_{2.5}$ concentration with removal behaviors, involving R-TY, R-T, and R-Y; $TPM_{2.5}$ represents the

10 simulated $PM_{2.5}$ concentration considering emission of anthropologic and CFB, which is correspond with the case of R0 (**Supplementary Fig. S2a**).

The sensitive study period was selected from 12:00 7[th] to 00:00 10[th]. **Fig. 8** displays the elevation contours and the horizontal distributions of $PM_{2.5}$ concentration with the effect of mountains. The results illustrate that the mountains had important impacts on regional $PM_{2.5}$ concentration, especially

the region along the foothill of mountains with a heavy pollution area, covering sampling sites of BJ, BD, SJZ and XT. Here, we summarized two categories of mountain effects, including: (1) In NCP, the Taihang Mountains is a major southwest-northeast mountain and the Yanshan Mountains is a major west-east mountain, when the wind blows from south to north or southeast to northwest, it is often blocked at the foothill of mountains, resulting in the high $PM_{2.5}$ loading (*mountain blocking effect*). (2)

When the prevailing winds are south-north or southeast-northwest, the Taihang Mountains act as a transmission guider oriented pollution accumulation along the foothill downwind areas (*mountain guiding effect*). Both effects act to prevent the pollutant plume to disperse toward west of mountains, causing accumulations of the air pollutants along the foothill of mountains. These two mountain effects are illustrated as the schematic pictures in **Supplementary Fig. S3**. The mountain effects were

quantified by the averaged horizontal distribution of $PM_{2.5}$ concentration.

**Fig. 9** displays the simulated $PM_{2.5}$ concentration due to the mountain effects ($RPM_{2.5}$), with the three cases (R-TY, R-T, and R-Y). The previous heavy pollution accumulation **(shown in Fig. 8)** along the foothill of mountains is significantly reduced, especially with the removal of Taihang Mountains (R-T, and RTY) (see **Fig. 9 a1** and **a2**). In these two cases, the pollution plumes dispersed westerly (see **Fig.**

**9 b1** and **b2**). It shows that the $PM_{2.5}$ concentrations increased 40-120 μg m[-3] in the western part of





Taihang Mountains, and reduced 20-60 μg m$^{-3}$ in NCP. The distribution of the reduced pollution plume shows a northeast band plume, indicating both the mountain blocking and guiding effects. With the case of removal the Yanshan Mountains (R-Y), the high PM$_{2.5}$ concentrations are still remained along the foothill of the Taihang Mountains (see **Fig. 9 a3**), but more pollutants are guided along the foothill

to the northeastern of NCP. Without the blocking effect of the Yanshan Mountains, the PM$_{2.5}$ concentrations increased 20-80 μg m$^{-3}$ in the northern part of the Yanshan Mountains, and decreased 10-60 μg m$^{-3}$ in the southern part of the Yanshan Mountains (see **Fig. 9 b3**).

In the foothill sampling sites (BJ, BD, SJZ and XT), the averaged PM$_{2.5}$ concentrations are reduced 56.0 μg m$^{-3}$ for the case of R-T, which is much higher than the case of R-Y (25.1 μg m$^{-3}$). For the other

non-foothill sites, the averaged reduction is 36.1 μg m$^{-3}$ for the case of R-T, which is also much higher than the case of R-Y (1.3 μg m$^{-3}$), suggesting that the Taihang Mountains have stronger effects than the Yanshan Mountains. The higher impacts in the foothill sampling sites than non-foothill sites are further demonstrated, including blocking and guiding effects of mountains on PM$_{2.5}$ pollutions in NCP.

### 5 Conclusions

In recent years, the NCP region, including the capital city of Beijing, has been suffering serious haze pollution problem, causing by multiply emissions. One of the causes is due to the CFB, which had not been carefully studied. In this study, we extracted a more detailed emission inventory of CFB based on the provincial statistical data and open crop fires captured by satellite (MODIS). A regional dynamical/chemical model (WRF-Chem) was applied to study the effect of CFB on the PM$_{2.5}$

concentrations in NCP. The results are summarized:

(1) In order to intensive performance of the model studies, the model simulations were intensive compared with the measured results in both PM$_{2.5}$ concentrations and meteorological parameters (wind speed, wind direction, and the PBLH). The WRF-Chem model reproduced the pollution episode with a good agreement with observations. The correlation coefficients (R) of simulated

and measured PM$_{2.5}$ concentration are 0.87 in NNCP and 0.80 in SNCP, and the related NMB are -14% in NNCP and -3% in SNCP. The simulated meteorological parameters (winds and PBLH) are also in good agreement with observations in both NNCP and SNCP.

(2) The CFB performs important contribution to PM$_{2.5}$ concentration and the maximum daily





averaged contributions are higher than 32% in both SNCP and NNCP. The contribution in SNCP is mainly due to local emission, whereas contribution in NNCP is largely resulted from regional transport.

(3)    The research domain includes two important areas. One is the north part of NCP (NNCP), including two mega cities (Beijing and Tianjin), where only few CFB occurred. Another is the south part of the NCP (SNCP), where substantial crop fires occurred during the haze episodes. Because of the haze problem in the capital city of China (Beijing), one of the main focuses is to study the long-range transport of CFB from SNCP to NNCP. This study shows that there are substantially long-transport of CFB plume from SNCP to NNCP. More importantly, the effect of CFB remains in a longer time in NNCP than in SNCP along the foothill of the Taihang Mountains, causing significant enhancement in Beijing in both time and magnitude.

(4)    Another major finding is that the mountains played significant roles in affecting the $PM_{2.5}$ pollution through the blocking effect and guiding effect. With the reduction of the topography altitudes, the dynamical conditions calculated from WRF-Chem change, which affect pollutions transport, especially along the foothill of mountains. The mountain blocking effect represents the phenomenon that pollutants are often blocked and then resulted in $PM_{2.5}$ accumulation at the foothill of mountains. The mountain guiding effect denotes the mountains act as a transmission guider oriented pollution accumulation along the foothill downwind areas.

This study suggests that the $PM_{2.5}$ emissions in the southern NCP should be significantly limited in order to reduce the occurrences of heavy haze events in NNCP region, including the Beijing City.

**Acknowledgement**

The PBL height and wind field data was obtained from the European Centre for Medium-Range Weather Forecasts (ECMWF) website (http://www.ecmwf.int/ products/data/). This work is supported by the National Natural Science Foundation of China (NSFC) under Grant Nos. 41275186 and 41430424, and the Open Fund of the State Key Laboratory of Loess and Quaternary Geology (SKLLQG1413). The National Center for Atmospheric Research is sponsored by the National Science Foundation.





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

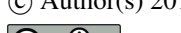

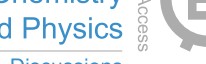
**Figure Captions**

**Figure 1** The study area, sampling sites and crop field burning. **(a)** The research domain and related provinces in China. **(b)** Topographical conditions of North China Plain. **(c)** Location of sampling sites and crop field burning captured by MODIS during the haze episodes. Green crosses indicate the measurement sites, and the crop field burning is shown by the pink dots.

**Figure 2** The **(a)** yearly and **(b)** monthly crop field burning observed by MODIS in the research domain during the year of 2006 to 2014.

**Figure 3** Crop field burning captured by MODIS with the background of MODIS real-time true color map (**Left**) and related CO emission (**Right**) on Oct. 6th and 7th.

**Figure 4** Regional averaged temporal variations in simulated and observed results of **(a)** $PM_{2.5}$ concentration, **(b)** wind speed, **(c)** wind direction and **(d)** PBLH over the regions of NNCP and SNCP.

**Figure 5** Hourly $PM_{2.5}$ concentration of observations (**obs**) and simulations (**sim-total** and **sim-anthro**) in **(a)** NNCP and **(b)** SNCP. **Sim-total** represents the simulations considering total emissions (anthropologic and crop field burning), whereas **sim-anthro** is the simulations only considering anthropologic emissions.

**Figure 6** Hourly contribution of crop field burning to $PM_{2.5}$ mass concentration ($CMPM_{2.5}$) **(a)** in SNCP, **(b)** in NNCP and **(c)** their comparison. The key point-in-local-times of T1 (23:00 6th), T2 (05:00 7th), T3 (20:00 7th) and T4 (19:00 8th) are signed with blue arrow.

**Figure 7** The distributions of $TPM_{2.5}$ and $CMPM_{2.5}$ of the key point-in-local-times of T1, T2, T3 and T4, which represent different pollution phase of emission from crop field burning to $PM_{2.5}$.

**Figure 8** The elevation contours and the averaged spatial distributions of horizontal winds and averaged $TPM_{2.5}$ during 12:00 7th to 00:00 10th. The point symbols of circles and squares were used to distinguish observation sites weather or not located at the foothill of mountains. Meanwhile, the 200-meter contour was highlighted with bold black line.

**Figure 9** The averaged spatial distribution of $PM_{2.5}$ concentration and horizontal winds during 12:00 7th to 00:00 10th. **(a)** Simulated $PM_{2.5}$ loading with removal behaviors ($RPM_{2.5}$), involving R-TY, R-T, and R-Y. **(b)** The related impacts of mountains to $PM_{2.5}$ ($IPM_{2.5}$), which represents the net effect of related mountains. The bold black lines were used to stress enclosing scope of each removal behavior.





Table 1. The average $PM_{2.5}$ concentration, wind direction and wind speed of the observations from 12:00 6[th] to 00:00 12[th]. The sampling sites located at the foot of mountains were emphasized with bold style.

| Region | Site | Longitude (°E) | Latitude (°N) | $PM_{2.5}$ ($\mu g/m^3$) | Wind-dir (°) | Wind-spd (m/s) |
|---|---|---|---|---|---|---|
| | **Beijing (BJ)** | 116.41 | 40.04 | **245.5** | 185.8 | 2.2 |
| | Langfang (LF) | 116.73 | 39.56 | 214.7 | 177.0 | 2.4 |
| | Tianjin (TJ) | 117.31 | 39.09 | 134.7 | 173.5 | 2.4 |
| | **Baoding (BD)** | 115.49 | 38.87 | **287.7** | 171.2 | 2.2 |
| | Cangzhou (CZ) | 116.87 | 38.31 | 117.3 | 166.6 | 2.5 |
| NNCP | | | | **200.0** | **174.8** | **2.35** |
| | **Shijiazhuang (SJZ)** | 114.49 | 38.04 | **257.9** | 175.2 | 2.0 |
| | Hengshui (HS) | 115.68 | 37.74 | 166.7 | 163.7 | 2.6 |
| | Dezhou (DZ) | 116.31 | 37.47 | 152.4 | 162.7 | 2.6 |
| | **Xingtai (XT)** | 114.50 | 37.09 | **320.1** | 198.1 | 2.3 |
| | Liaocheng (LC) | 116.00 | 36.46 | 139.7 | 158.4 | 2.6 |
| | Hezhe (HZ) | 115.46 | 35.26 | 105.0 | 138.9 | 2.4 |
| | Zhengzhou (ZZ) | 113.66 | 34.79 | 146.9 | 159.2 | 2.4 |
| SNCP | | | | **184.1** | **165.2** | **2.42** |

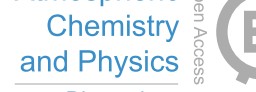

Table 2. The gaseous and particulate species emission factors (g/kg) and scales to convert NMOC emissions (kg day$^{-1}$) to WRF/Chem chemical species (moles-species day$^{-1}$) from crop field burning. The detailed chemical species are described by Stockwell et al. (1990).

| Gaseous species | | | | | | | Particulate species | | |
|---|---|---|---|---|---|---|---|---|---|
| CO[1] | NOx[1] | NO[1] | NO$_2$[2] | SO$_2$[3] | NH3[1] | NMOC[1] | OC[3] | BC[3] | PM$_{2.5}$[1] |
| 111 | 3.5 | 1.7 | 3.9 | 0.4 | 2.3 | 57 | 3.3 | 0.69 | 5.8 |

| Chemical-compounds-to-NMOC scales[1,2] | | | | | | | | | | | |
|---|---|---|---|---|---|---|---|---|---|---|---|
| ETH | HC3 | HC5 | OL2 | OLT | OLI | TOL | CSL | HCHO | ALD | KET | ORA2 | ISO |
| 0.43 | 0.73 | 0.07 | 1.09 | 0.27 | 0.20 | 1.07 | 0.49 | 1.84 | 3.05 | 0.83 | 2.19 | 0.60 |

[1] Andreae and Merlet (2001)

[2] Wiedinmyer et al., (2001)

[3] Akagi et al., (2011)





Table 3. The fire counts of crop field burning detected by the MODIS in the provinces over NCP during the haze episode (from Oct. 6[th] to 11[th], 2014)

| Province | 6-Oct | 7-Oct | 8-Oct | 9-Oct | 10-Oct | 11-Oct | Percentage |
|---|---|---|---|---|---|---|---|
| Beijing | 0 | 0 | 0 | 0 | 0 | 0 | 0% |
| Tianjin | 0 | 0 | 0 | 0 | 0 | 0 | 0% |
| Hebei | 60 | 11 | 14 | 1 | 5 | 6 | 10% |
| Henan | **370** | **104** | 59 | 18 | 19 | 23 | **61%** |
| Shandong | **100** | **54** | 9 | 9 | 32 | 7 | **22%** |
| Anhui | 6 | 6 | 20 | 0 | 10 | 3 | 5% |
| Shanxi | 3 | 0 | 0 | 3 | 4 | 1 | 1% |
| Jiangsu | 4 | 3 | 5 | 0 | 3 | 1 | 2% |
| **Percentage** | **56%** | **18%** | 11% | 3% | 8% | 4% | 100% |





Table 4. The emissions (Gg/day) of gaseous and particulate species from crop field burning on Oct. 6[th] and Oct. 7[th] in NCP region, including the provinces of Beijing, Tianjin, Hebei, Henan, Shandong.

| Time | Province | CO | NOx | NO | NO2 | NMOC | SO2 | NH3 | PM2.5 | OC | BC |
|------|----------|-----|-----|-----|-----|------|-----|-----|-------|-----|-----|
| 6-Oct | Beijing | 0.00 | 0.00 | 0.00 | 0.00 | 0.00 | 0.00 | 0.00 | 0.00 | 0.00 | 0.00 |
| | Tianjin | 0.00 | 0.00 | 0.00 | 0.00 | 0.00 | 0.00 | 0.00 | 0.00 | 0.00 | 0.00 |
| | Hebei | 10.73 | 0.34 | 0.16 | 0.38 | 5.51 | 0.04 | 0.22 | 0.56 | 0.32 | 0.07 |
| | Henan | 75.87 | 2.39 | 1.16 | 2.67 | 38.96 | 0.27 | 1.57 | 3.96 | 2.26 | 0.47 |
| | Shandong | 17.35 | 0.55 | 0.27 | 0.61 | 8.91 | 0.06 | 0.36 | 0.91 | 0.52 | 0.11 |
| | **Total** | **103.9** | **3.3** | **1.6** | **3.7** | **53.4** | **0.4** | **2.2** | **5.4** | **3.1** | **0.6** |
| 7-Oct | Beijing | 0.00 | 0.00 | 0.00 | 0.00 | 0.00 | 0.00 | 0.00 | 0.00 | 0.00 | 0.00 |
| | Tianjin | 0.00 | 0.00 | 0.00 | 0.00 | 0.00 | 0.00 | 0.00 | 0.00 | 0.00 | 0.00 |
| | Hebei | 1.97 | 0.06 | 0.03 | 0.07 | 1.01 | 0.01 | 0.04 | 0.10 | 0.06 | 0.01 |
| | Henan | 21.32 | 0.67 | 0.33 | 0.75 | 10.95 | 0.08 | 0.44 | 1.11 | 0.63 | 0.13 |
| | Shandong | 9.37 | 0.30 | 0.14 | 0.33 | 4.81 | 0.03 | 0.19 | 0.49 | 0.28 | 0.06 |
| | **Total** | **32.7** | **1.0** | **0.5** | **1.1** | **16.8** | **0.1** | **0.7** | **1.7** | **1.0** | **0.2** |





Table 5. Averaged contribution ration of crop field burning to $PM_{2.5}$ concentration

| Region | 6-Oct. | 7-Oct. | 8-Oct. | 9-Oct. | 10-Oct. | 11-Oct. |
|--------|--------|--------|--------|--------|---------|---------|
| NNCP | 4% | **32%** | **10%** | 3% | 2% | 4% |
| SNCP | **35%** | **17%** | 6% | 3% | 1% | 1% |





Table 6. The regional statistical results of crop field burning contribution in mass concentration of $PM_{2.5}$ (µg) for the four time-points of T1 (23:00 6[th]), T2 (05:00 7[th]), T3 (20:00 7[th]) and T4 (19:00 8[th]).

| Region | T1 | T2 | T3 | T4 |
|---|---|---|---|---|
| **Mean value of** $CMPM_{2.5}$ | | | | |
| NNCP | 7.7 | 37.0 | 52.8 | 14.2 |
| SNCP | 77.0 | 38.7 | 21.6 | 9.8 |
| **Maximum value of** $CMPM_{2.5}$ | | | | |
| NNCP | 68.2 | 154.2 | 127.4 | 55.7 |
| SNCP | 340.5 | 170.8 | 81.5 | 126 |
| **First quartile (Q1) of** $CMPM_{2.5}$ | | | | |
| NNCP | 0.2 | 9.5 | 28.4 | 5.5 |
| SNCP | 23.2 | 10.6 | 4.6 | 1.8 |
| **Third quartile (Q3) of** $CMPM_{2.5}$ | | | | |
| NNCP | 10.1 | 52.2 | 71.7 | 21.1 |
| SNCP | 109.3 | 57.6 | 35.6 | 8.6 |

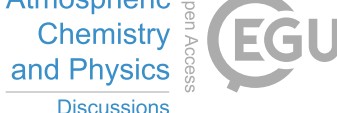



Figure 1



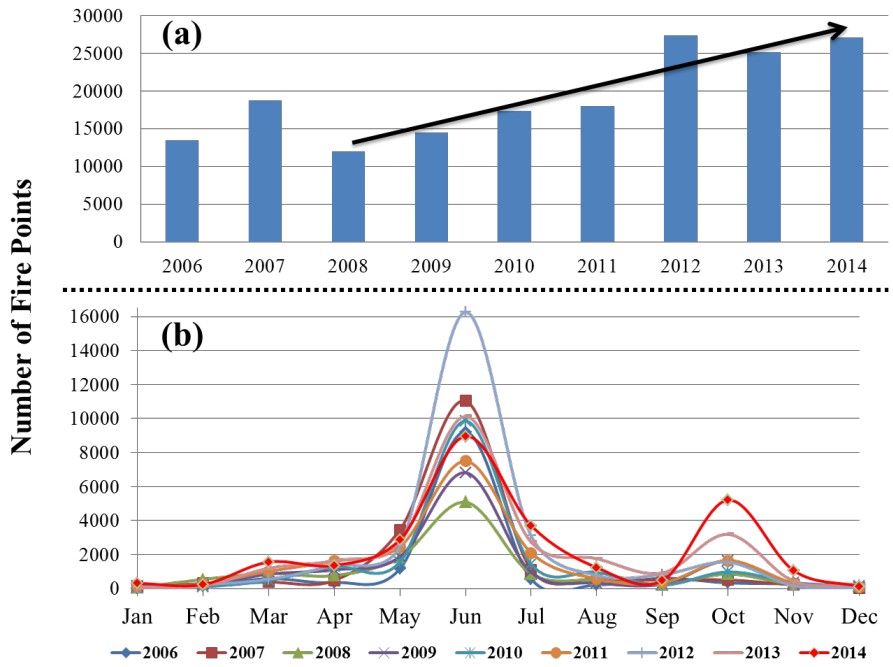

Figure 2



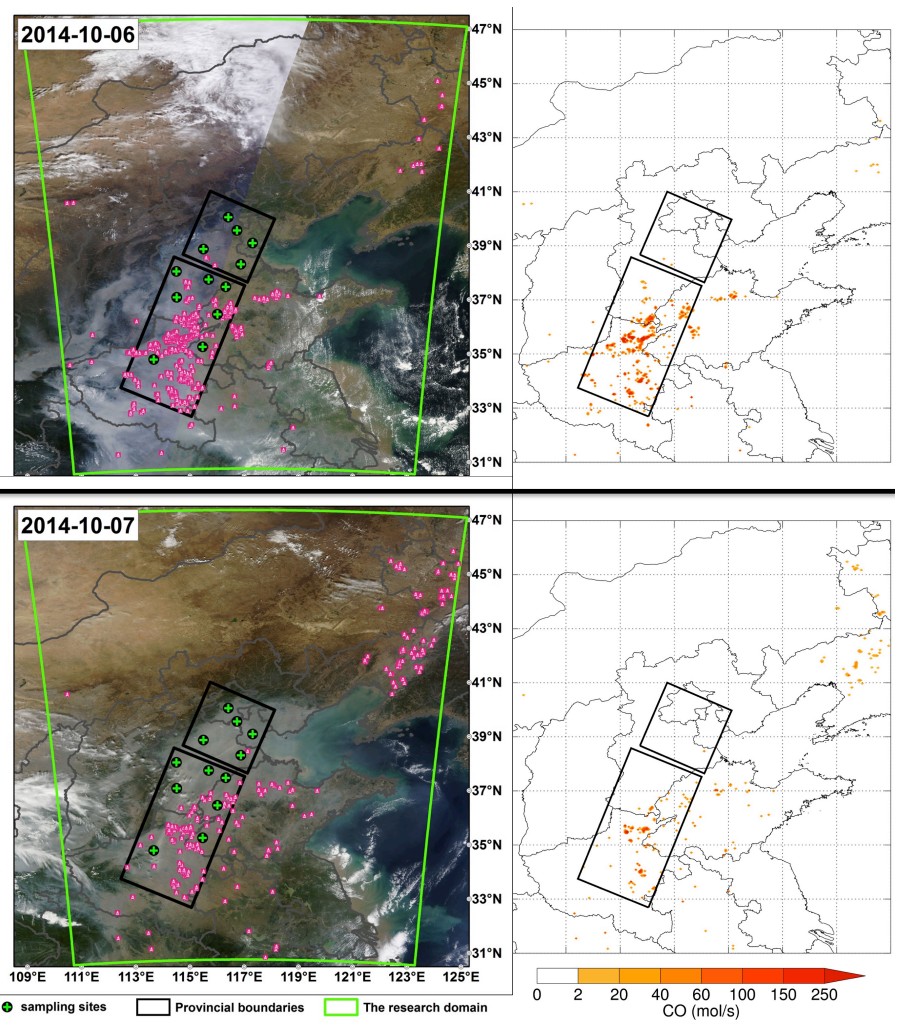

Figure 3





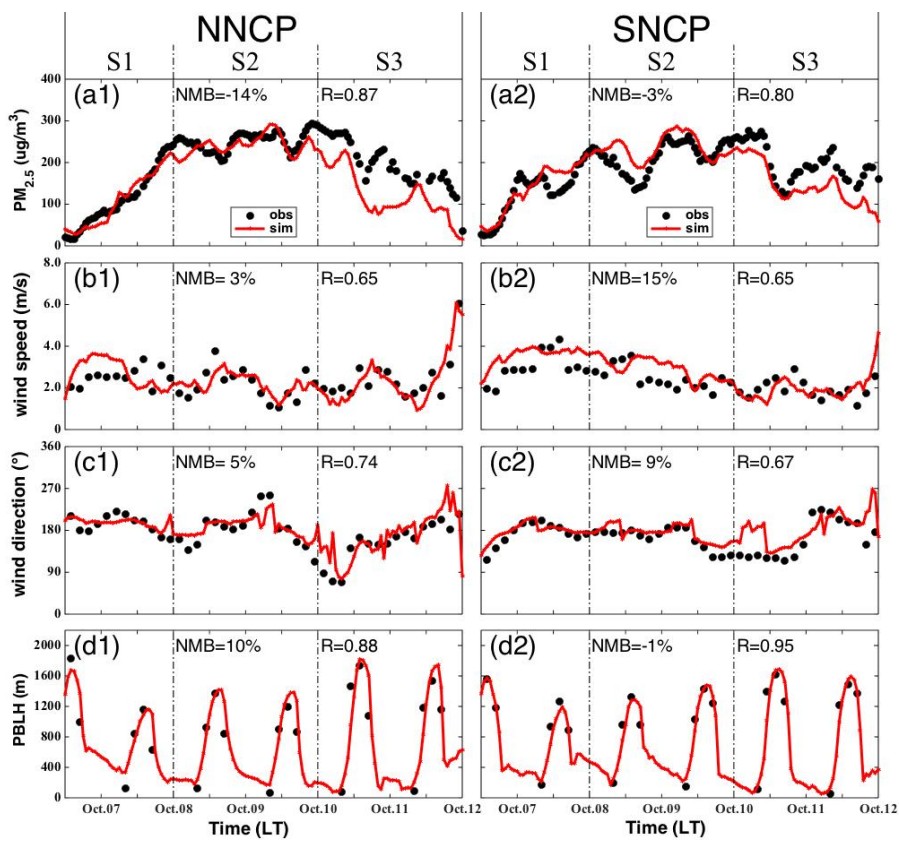

Figure 4





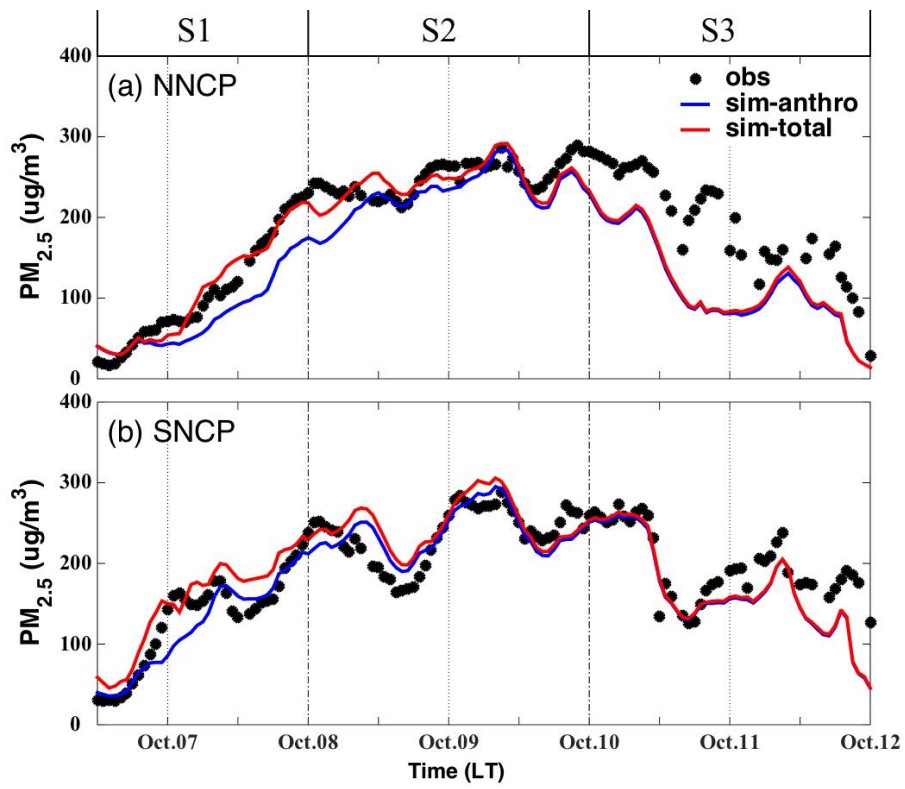

Figure 5





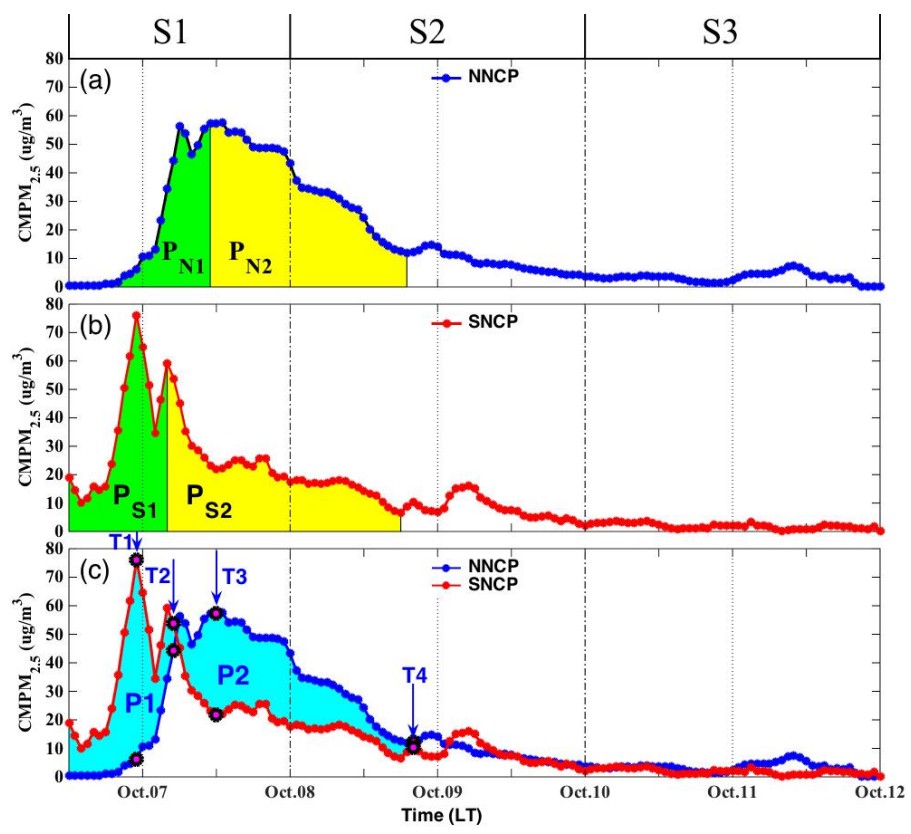

Figure 6



Figure 7





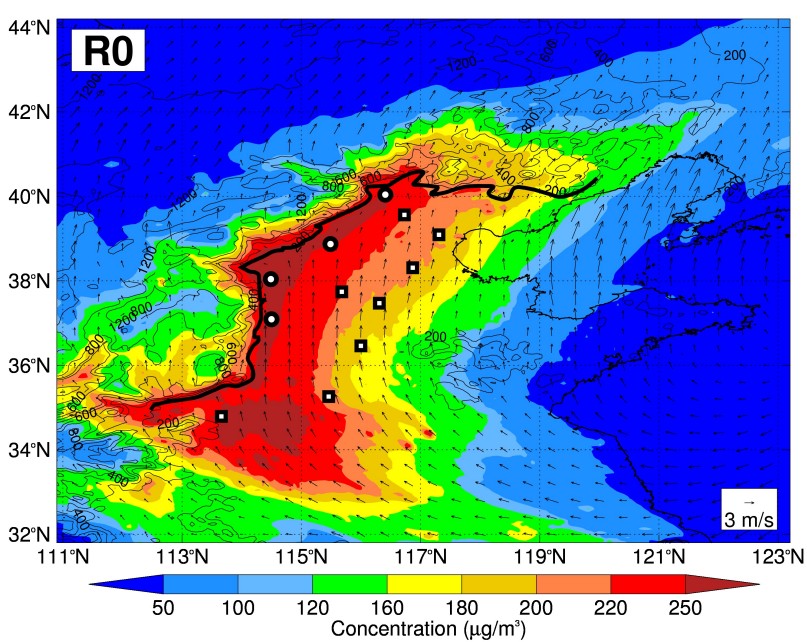

Figure 8

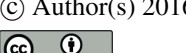

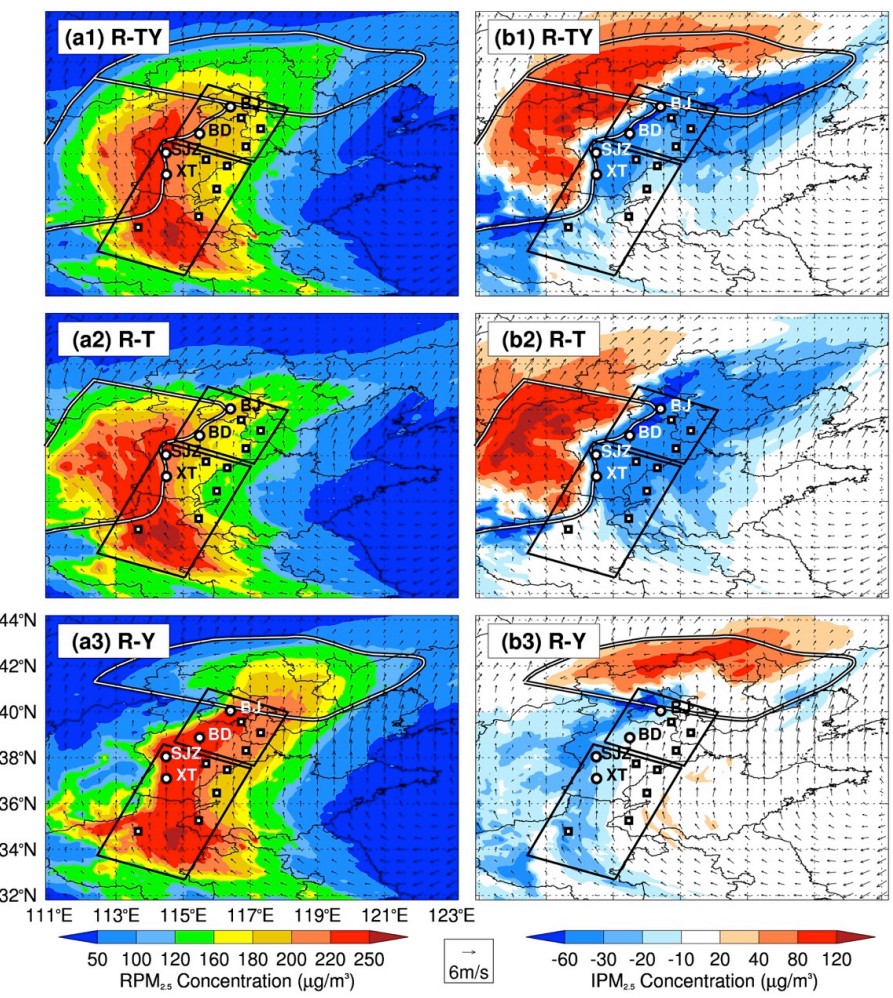

Figure 9

