# Peer review of "Impact of crop field burning and mountains on heavy haze in the North China Plain: A case study"

_Atmospheric Chemistry and Physics, 2016_

## Referee Comment (RC1) · Anonymous Referee #1 · 13 Apr 2016

The paper by Long X. et al. assesses the impact of crop field burning (CFB) and topography on air quality in North China Plain. The contribution of crop field burning is quantified. This analysis would have a substantial impact on policy. However, I think the impact of CFB and mountainous topography are two distinct impact sources. Please justify the reason to address these two distinct impacts in this single study.

Introduction. Page 3 Line 17, "…it is lack of study for the quantitative effect…" I do not think it is appropriate to claim this without justification. A number of source apportionment studies have quantilfied the contribution of biomass burning in Beijing with modeling approach1, 2,3. A more comprehensive review of previous studies should be summarized in this part. It should be also noted in the manuscript on what the novelty of this study is. Also, Summary of references on biomass burning emissions and the influence of mountains in NCP on air pollution is needed in introduction part as well.

Further work on modeling evaluation and validation of model are needed. Page 5 Line 20 states that the aerosol module from CMAQv4.6, released in 2006 is used in this study. Could the authors explain why to choose version 4.6 instead the latest version of CMAQ? It has been accounted by Baek, et al. 4 that the simulated OM tends to be underestimated due to the uncertainty in secondary organic aerosols mechanism. However, Figure 4 shows that simulated and observed PM2.5 mass concentration matches well. So is it possible to evaluate the model with PM2.5 species mass concentration and their precursor mass concentrations?

Page 9 Line 10 states that "strong southerly wind, with mean wind speed of 2.5 (2.7) m s-1 in NNCP and 3.0 (3.6) m s-1 in SNCP" to illustrate that "The pollution is continuously transported from SNCP to NNCP". It's not strong enough to get such conclusion. Trajectory analysis and wind speed profile analysis should be included.

In Section 4.5, it is written "The differences between the simulations with or without mountains showed the net effect of the topography on PM2.5 concentration". I wonder if it is appropriate to make this assumption for several reasons. First, the impact of topography is complicated. I am not sure if it is good to represent it just as "reduced to the averaged altitude". Second, the NCEP FNL Operational Global Analysis data is employed as the initial meteorological condition. It means the initial condition is "real" (with moutains) in all scenarios. The spin-up time is only 12 hours. I think the spin-up time is not long enough to get balanced. Also I wonder if any nudging method is used in this study (it should be explained in method part). If so, the contribution might be changed due to the nudging. Third, the domain is not large enough to ignore the impacts of "real" bounadary condition. The mountainous topography may change the large-scale circulation.

1. Yao, L.; Yang, L. X.; Yuan, Q.; Yan, C.; Dong, C.; Meng, C. P.; Sui, X.; Yang, F.; Lu, Y. L.; Wang, W. X., Sources apportionment of PM2.5 in a background site in the North China Plain. Science of the Total Environment 2016, 541, 590-598. 2. Mukai, S.; Yasumoto, M.; Nakata, M., Estimation of Biomass Burning Influence on

Air Pollution around Beijing from an Aerosol Retrieval Model. Scientific World Journal 2014. 3. Cheng, Z.; Wang, S.; Fu, X.; Watson, J. G.; Jiang, J.; Fu, Q.; Chen, C.; Xu, B.; Yu, J.; Chow, J. C.; Hao, J., Impact of biomass burning on haze pollution in the Yangtze River delta, China: a case study in summer 2011. Atmospheric Chemistry and Physics 2014, 14, (9), 4573-4585. 4. Baek, J.; Hu, Y.; Odman, M. T.; Russell, A. G., Modeling secondary organic aerosol in CMAQ using multigenerational oxidation of semi-volatile organic compounds. Journal of Geophysical Research: Atmospheres 2011, 116, (D22), D22204 (12 pp.)-D22204 (12 pp.).

---

## Referee Comment (RC2) · Anonymous Referee #2 · 8 May 2016

This paper conducts a numerical study to investigate the impact of crop field burning and topography on haze pollution in the North China Plain. This is an interesting study and potentially will be useful for air quality management in this region. However, there are several important points need to be appropriately addressed before it can be accepted for publishing in Atmospheric Chemistry and Physics.

Main comments:

1) The impacts of biomass burning and topography on air quality in the North China Plain certainly are interesting topics for policy makers. However, because modeling results are sensitive from case to case, from policy perspectives, such kind of study should be conducted for a longer period. For the crop filed burning, this paper only conducted a case study for one week in October. According to Figure 2 of this paper

and other previous studies in China, the most intensive period of biomass burning in the eastern China is June. It's a little bit strange that the authors selected a case in October. In fact, from the results presented in Figure 5, it is quite clear that in this case the crop field burning activities didn't play an important role on air quality in both NNCP and SNCP if it was compared with an overall broad peak of anthropogenic pollution. For suck kind of non-typical case, I don't know whether it is meaningful to make many statistics to compare the relative contributions in several tables (e.g. Tables 3-6). With such a short-term case, I would like suggest conducting more in-depth analysis to understand specific scientific questions other than a calculation of numbers.

2) In the second part of this paper, the authors conducted an interesting numerical experiment by removing topography in specific regions in WRF-Chem model. However, such kind of treatment may cause some inconsistency in the initial conditions of meteorological parameters and the terrain data, which need more spin-up time because WRF uses a terrain-following vertical coordinate. However, according to the modeling description of this paper, the spin-up time for the WRF-Chem simulation is only 12 hours. Since the authors aim to give a quantitative understanding of the topographic effects, a longer spin-up time, for example several days, is needed. In addition, same as the Comment #1, as a case study for several days, the quantitative results here will have large uncertainty for policy makers. I would like suggest giving a more in-depth discussion by touching some scientific questions related to mountain, such as the impact of mountain-valley breezes on the accumulation of air pollutants etc.

3) The authors didn't give appropriate literature review for the both topics of biomass burning and topographic effect. In the model description part, the authors gave too many (more than 15) unnecessary references related to some common model schemes in WRF-Chem with some of them published several decades ago. However, in the main results part (Sect. 4), only two references (Cao et al., 2008 & Huang et al., 2012) are cited in the first paragraph but there is a lack of some comparisons of the results and conclusions with previous works done by other scientists for similar topics

in the same regions or in other regions of the world.

---

## Author Response (AR1)

May 17, 2016

Dear Editor,

We have received the comments from the two reviewers of the manuscript. Below are our responses and the revisions that we have made in the manuscript.

Thank you for your efforts on this manuscript. We look forward to hearing from you.

Best Regards,

Xuexi Tie

*Response to Referee #1*

We thank the Referee for the careful reading of the manuscript and helpful comments. According to the suggestions of the referee, the comments have been carefully addressed, and the paper is carefully revised. We believe that the revised paper has been significantly improved after addressing the comments of the referee. We respond to each specific comment below. The original comments by the Referee are shown in bold italics. Our reply is shown in blue.

*General comments:*

*1. The paper by Long X. et al. assesses the impact of crop field burning (CFB) and topography on air quality in North China Plain. The contribution of crop field burning is quantified. This analysis would have a substantial impact on policy.*

We thank the referee for the careful reading and the valuable comments that helped improving our paper.

*2. However, I think the impact of CFB and mountainous topography are two distinct impact sources. Please justify the reason to address these two distinct impacts in this single study.*

We thank the referee for the thoughtful comment. We added these two impacts together because these two effects are related to each other by the following reasons.

a. One goal of the study is to analyze the effect of CFB on aerosol pollution in the northern NCP (NNCP), where the capital city of China (Beijing) locates. Whereas the major CFB occurred in the southern NCP (SNCP). As shown in the analysis, with the prevailing southerly winds, the regional transport plays important roles to transport CFB pollutants from SNCP to NNCP.

b. As shown by the measurements, the mountains play important roles for the northward transport, and cause the accumulation of the aerosol pollutants at the foothill of the mountains. By considering the above reasons, it is important to add these two important effects together in the analysis. We added some statements in the revised paper.

Line 99-104, "Thereafter we analyzed the regional transport of CFB emissions from

SNCP to NNCP driven by prevailing southerly winds. Under the continuously southerly wind condition, the mountains play important roles for the northward transport, and cause the accumulation of the aerosol pollutants at the foothill of the mountains. We also analyzed the impact of mountains (especially the Taihang Mountains and the Yanshan Mountains) on the air pollution transport."

Line 147-149, "Considering the continuously southerly winds and the topographic conditions, we studied the impact of the mountains on the air pollution transport."

Line 351-355, "Indeed, the CFB pollution plume go through a long-range transport to NNCP can cause an obvious increase to $PM_{2.5}$ concentration, with the maximum daily average contribution of 32% (**Table 5**). Such a high transported contribution indicates that the CFB is not only one of the significant local pollution sources, but also a considerable regional pollution source."

**Detailed Comments**

*1. Introduction. Page 3 Line 17, " . . .it is lack of study for the quantitative effect . . ." I do not think it is appropriate to claim this without justification. A number of source apportionment studies have quantified the contribution of biomass burning in Beijing with modeling approach[1,2,3]. A more comprehensive review of previous studies should be summarized in this part.*

To response the Referee's comments, we modified and added a summary of previous studies.

Line 59-67, "Numerous studies have quantified the contribution of biomass burning and CFB to PM pollution in China. According to Yao et al. (2016), Cheng et al. (2013), Wang et al. (2009; 2007) and Song et al. (2007), biomass burning has important impacts on the ambient $PM_{2.5}$ concentrations (15-24% in Beijing and 4-19% in Guangzhou). Yan et al. (2010) captured a heavy pollution with $PM_{10}$ concentrations higher than 350 $\mu g\ m^{-3}$ in some CFB locations. It is reported that CFB may contribute more than 30% of the $PM_{10}$ increase during CFB incidents (Zhu et al., 2012; Zha et al. 2013;Su et al., 2012). Cheng et al. (2014) report a summer case that CFB contributed 37% of $PM_{2.5}$ concentrations in the Yangtze River delta."

Line 67-75, "The impact of CFB to air quality is continental and regional. Air quality in China is influenced by the CFB occurred in Southeast Asia and on the Indian Peninsula (Qin et al., 2006). Mukai et al. (2014) have reported that CFB emissions in Southeast Asia contribute the carbonaceous aerosols in Beijing. Within China, the inter-province transported air pollutants emitted from CFB significantly affect regional PM levels and air quality (Cheng et al., 2014;Zhu et al., 2012). For Beijing, the smoke particles from CFB are expected to be one of the major components (Wang et al., 2014;Cheng et al., 2013), though the percentage of transported sources are seldom specified (Zhang et al., 2016)."

*2. It should be also noted in the manuscript on what the novelty of this study is.*

Thanks. The novelty of this study is to use multiply methods to quantify the impacts of a serious CFB incident on the aerosol pollution in regional scale. The main methods and conclusions include (1) using satellite data to generate the CFB emission inventory with higher temporal and spatial resolution, (2) using WRF-CHEM model to study the regional transport from the burning region (SNCP) to the NNCP (where the capital city of Beijing locates), and (3) quantifying the effect of the mountains on the accumulation of pollutants at the foothill of the mountains. The combination of the multiply methods provides a better understanding of the effect of CFB on the regional air pollution. We modified and added explicit statements in the revised paper.

Line 32-34, "This study suggests that the prohibition of CFB should be strict not just in or around Beijing, but also on the ulterior crop growth areas of SNCP."

Line 449-456, "In recent years, the NCP region, including the capital city of Beijing, has been suffering serious haze pollution problem, especially in winter and summer. Most studies concerned about the intense secondary formation, huge regional transport of pollutants, stationary meteorological conditions and large local emission. In autumn, CFB and movement of wind based on large scale topography are important in NCP, whereas the percentage of transported CFB emission sources are seldom specified. This is probably resulted from the contingency of CFB activities during harvest period and the limitation of temporal resolution of CFB emission inventories."

, "A more detailed CFB emission inventory was generated in NCP. The daily CFB emissions were estimated depending on CFB activities captured by MODIS. Plenty of pollutants emitted from SNCP on Oct. $6^{th}$ and $7^{th}$, producing plenty of $PM_{2.5}$ pollution, but few in NNCP during the entire haze period."

, "Another major finding is that the mountains, surrounding the NCP in the north and west, play significant roles in enhancing the $PM_{2.5}$ pollution in NNCP through the blocking effect. The mountains block and redirect the airflows, causing the pollution accumulation along the foothill of mountains. The Taihang Mountains had greater impacts on $PM_{2.5}$ concentration than the Yanshan Mountains.

On account of various factors, such as pollutant long-range transport and pollutant accumulation caused by mountain effects, the prohibition of CFB should be strict not just in or around Beijing, but also on the ulterior crop growth areas of SNCP. Other $PM_{2.5}$ emissions in the SNCP should be significantly limited in order to reduce the occurrences of heavy haze events in NNCP region, including the Beijing City."

3. *Also, Summary of references on biomass burning emissions and the influence of mountains in NCP on air pollution is needed in introduction part as well.*

a. A comprehensive summary of biomass burning emission has been added in the revised paper.

, "Crop residue resources in China rank the first in the world, accounting for 17.3% of the global production (Bi et al., 2010), and increasing with the average annual proportion of 4% (Hong et al., 2015;Zhao et al., 2010). Compared with other approaches, crop field burning (CFB) is the most effective and less expensive to remove residues. The national annual average proportion of CFB to total residues is about 11-25%(Cao et al., 2008;Hao and Liu, 1994;Streets et al., 2003;Wang and Zhang, 2008;Zhao et al., 2010). Large numbers of annual CFB occur in China (Zhang et al., 2015; Yan et al., 2006), especially during the post-harvest seasons (Zhang et al., 2016;Shi et al., 2014;Cao et al., 2008)."

, "However, CFB have adverse impacts on traffic conditions and ecology environments (Shi et al., 2014;Zhang, 2009), and release plenty of pollutants, such as CO, $SO_2$, VOC, NOx and $PM_{2.5}$ (Koppmann et al., 2005;Li et al., 2008). According to Guan et al. (2014) and Lu et al. (2011), annual CFB contribute about 13% of the total particulate matter (PM) emissions in China (Zhang et al., 2016). And it is more prominent during the harvest periods due to its strong seasonal dependence. Numerous studies have quantified the contribution of biomass burning and CFB to PM pollution in China. According to Yao et al. (2016), Cheng et al. (2013), Wang et al. (2009; 2007) and Song et al. (2007), biomass burning has important impacts on the ambient $PM_{2.5}$ concentrations (15-24% in Beijing and 4-19% in Guangzhou). Yan et al. (2010) captured a heavy pollution with $PM_{10}$ concentrations higher than 350 $\mu g\ m^{-3}$ in some CFB locations. It is reported that CFB may contribute more than 30% of the $PM_{10}$ increase during CFB incidents (Zhu et al., 2012; Zha et al. 2013;Su et al., 2012). Cheng et al. (2014) report a summer case that CFB contributed 37% of $PM_{2.5}$ concentrations in the Yangtze River delta."

b. We modified and added explicit statements of provincial CFB emission inventory processing in Line 192-197 and Line 210-221. And we updated the provincial statistical data and related results. The detailed results and related references were added in supplementary data of Table S1, Table S2 and Table S3.

Line 192-197, "This situation may be resulted from the limitation of local enforcement of regulation despite CFB have already been banned (Zhang and Cao, 2015;Shi et al., 2014). The CFB have a seasonal pattern due to the post-harvest activities with two distinct peaks in summer and autumn, especially in June (33-59%) and October (6-19%) (**Fig. 2b**). The strong seasonal dependence character suggests that the CFB emissions during October are much larger than annual averages."

Line 210-221, "where i stands for each province and k for different crop species of rice, corn and wheat. $E_{i,co}$ stands for CO emission from CFB of *i-th* province in gigagrams

[Gg]. $P_{i,k}$ is the yield of crop in Gg. $F_i$ is the proportion of residues burned in the field. $D_k$ is the dry fraction of crop residue (dry matter). $R_k$ is the residue-to-crop ratio (dry matter). $CE_k$ is the combustion efficiency and $EF_{co}$ is the emission factors of CFB. The $P_{i,k}$ values were taken from an official statistical yearbook (NBS, 2015) (**Table S1**), and the $F_i$ on a provincial basis were taken from Wang and Zhang (2008) and Zhang Yisheng (Unpublished doctor thesis-in Chinese) (**Table S1**). The parameters of $D_k$, $R_k$, and $CE_k$ are listed in **Table S2**. The $EF_{co}$ from CFB was summarized range from 52 to 141 g kg$^{-1}$ in China (**Table S3**). In this study, we used 111 g kg$^{-1}$ as the average $EF_{co}$ of crop residue, which was used to estimate the emissions from global open burning (Wiedinmyer et al., 2011)."

c. The influence of mountains in NCP on air pollution is added in ***Introduction.***

Line 80-91, "Yanshan and Taihang Mountains surround the NCP in the north and west (**Fig. 1c**). Such topography affects air pollution though PBL in complex ways (Miao et al., 2015b;Sun et al., 2013;Liu et al., 2009). Hu et al. (2014) have reported that the Loess Plateau and NCP result in a mountain-plains solenoid circulation, exacerbating air pollution over NCP. Chen et al. (2009) have founded that a mountain chimney effect is dominated by mountain-valley breeze, enhancing the surface air pollution in Beijing. The mountain-plain breeze develops frequently in Beijing and may play important roles in modulating the local air quality (Miao et al., 2015b;Hu et al., 2014;Chen et al., 2009). Miao et al. (2015a) founded that the mountains played a significant role in the sea-land aerosol circulation and the pollutants could be transported and accumulated in the NCP areas along the mountains, which is treated as the blocking effect (Zhao et al., 2015)."

4. *Further work on modeling evaluation and validation of model are needed. Page 5 Line 20 states that the aerosol module from CMAQv4.6, released in 2006 is used in this study. Could the authors explain why to choose version 4.6 instead the latest version of CMAQ? It has been accounted by Baek, et al. 4 that the simulated OM tends to be underestimated due to the uncertainty in secondary organic aerosols mechanism. However, Figure 4 shows that*

*simulated and observed PM2.5 mass concentration matches well. So is it possible to evaluate the model with PM2.5 species mass concentration and their precursor mass concentrations?*

In order to response the referee's comments, we added several revisions. We believe that these revisions help to better evaluate the model result.

a. A more detailed model description was added in **Section 3.1 Model description**:

Line 153-160, "The specific version of WRF-CHEM model is developed by Li et al. (2010; 2011; 2012), with a new flexible gas phase chemical module and the CMAQ (version 4.6) aerosol module developed by US EPA (Binkowski and Roselle, 2003). The wet deposition follows the CMAQ method and the dry deposition is parameterized following Wesely (1989). The photolysis rates are calculated using the FTUV (Li et al., 2005;Tie et al., 2003), in which the impacts of aerosols and clouds on the photochemistry are considered (Li et al., 2011)."

Line 162-166, "Meanwhile, the ISORROPIA Version 1.7 (http://nenes.eas.gatech.edu/ISORROPIA/) is utilized to simulate the inorganic aerosols, which is primarily used to predict the thermodynamic equilibrium between the ammonia-sulfate-nitrate-chloride-water aerosols and their gas phase precursors of $H_2SO_4$-$HNO_3$-$NH_3$-HCl-water vapor."

Line 184-185, "The biogenic emissions are calculated on-line with the WRF-CHEM model using the MEGAN model (Guenther, 2006)."

b. The explanation of the statistical characteristics of the evaluation is added.

, **"The simulations are overall lower than the observations with NMB of -12% in NNCP and -7% in SNCP. Considering the high average $PM_{2.5}$ concentration with 200.0 μg m$^{-3}$ in NNCP and 184.1 μg m$^{-3}$ in SNCP, obvious underestimates exist with the overall concentrations of 24.0 μg m$^{-3}$ in NNCP and 12.9 μg m$^{-3}$ in SNPC. This may be related to the CMAQ (version 4.6) aerosol module, which is likely to underestimated OM due to the uncertainty in secondary organic aerosols mechanism (Baek et al., 2011). Meanwhile, the underestimates are also related to the negative bias in S3, which may be related to cloud contamination (Fig. S1)."**

c. More evaluations of model calculation, such as $NO_2$ and $O_3$, were added **in the Section 4.2 Statistical characteristics of the evaluation**. A new figure of the comparison between the calculated and measured $NO_2$ and $O_3$ is also added in the section (Fig. 4b and 4c), and related descriptions were modified.

, **"In order to evaluate the model performance, the model simulations were compared with the measured results in both species concentrations ($PM_{2.5}$, $O_3$ and $NO_2$) and meteorological parameters (wind speed, wind direction and PBLH)."**

, **"Figure 4** shows the measured and calculated temporal variations of regional average species concentrations, including $PM_{2.5}$, $O_3$ and $NO_2$. The WRF-CHEM model reproduced the pollution episode well, with a good agreement with observations. The correlation coefficients (R) of simulated and measured $PM_{2.5}$ concentrations are 0.88 in both NNCP and SNCP (**Fig. 4a**)."

, **"The simulations of $O_3$ and $NO_2$ are also agree well with observations, with R greater than 0.77 and absolute NMB lower than 17% (Fig. 4b and 4c)"**

d. Pattern comparisons of simulated vs. observed near-surface $PM_{2.5}$ concentrations were added in Fig. 9 and Fig.10.

Line 370-372, "The pattern comparisons between simulated and observed near-surface PM$_{2.5}$ concentrations (*TPM$_{2.5}$*) perform well (**Fig. 9 Left Panels**)."

Line 767-770, "**Figure 9 …** Left panels also show the pattern comparisons of simulated vs. observed near-surface PM$_{2.5}$ concentrations (*TPM$_{2.5}$*), with PM$_{2.5}$ observations of colored circles. Black arrows denote simulated surface winds."

Line 414-415, "exhibiting a good performance of the pattern comparisons between simulated and observed near-surface PM$_{2.5}$ concentrations."

Line 771-775, "**Figure 10** The elevation contours and the pattern comparisons of simulated vs. observed near-surface PM$_{2.5}$ concentrations from 12:00 7$^{th}$ to 00:00 10$^{th}$. Colored circles: PM$_{2.5}$ observations of foothill sites; Colored squares: PM$_{2.5}$ observations of non-foothill sites; Black arrows: simulated surface winds. The 200-meter contour was highlighted with bold black line."

5. *Page 9 Line 10 states that "strong southerly wind, with mean wind speed of 2.5 (2.7) m s-1 in NNCP and 3.0 (3.6) m s-1 in SNCP" to illustrate that "The pollution is continuously transported from SNCP to NNCP". It's not strong enough to get such conclusion. Trajectory analysis and wind speed profile analysis should be included.*

According to the referee's suggestions, we added a backward trajectory analysis and wind speed profile analysis **in the Section 4.3 Characteristics of the heavy pollution events.** A new figure of the backward trajectory analysis during S1 was added in the revised paper. As shown in Fig. 6, the prevailing wind during the analysis period (S1) is continuously from south to north.

Line 296-301, "The backward trajectories, with the HYSPLIT model online version, of

BJ, TJ and BD during S1 reflected how the CFB influenced the NNCP region (**Fig. 6**). The air mass mainly came from the south, originating from the SNCP region. The pollutants are continuously transported from SNCP to NNCP, leading to pollutants accumulation in NNCP…"

***6. In Section 4.5, it is written "The differences between the simulations with or without mountains showed the net effect of the topography on PM2.5 concentration". I wonder if it is appropriate to make this assumption for several reasons. First, the impact of topography is complicated. I am not sure if it is good to represent it just as "reduced to the averaged altitude". Second, the NCEP FNL Operational Global Analysis data is employed as the initial meteorological condition. It means the initial condition is "real" (with mountains) in all scenarios.***

a. Thanks for the suggestion. As an online model, the reduction of the topography in WRF-CHEM can lead to dynamical changes, such as the wind speeds at the foothill of the mountains. We agree with the referee that there are some shortcomings of the method, and we modified the text to point out these shortcomings **in the Section. 4.5 Impact of mountains**.

Line 396-405, **"**In this study, we utilized the differences between the simulations with or without mountains to represent the effect of the topography on $PM_{2.5}$ concentration, which were calculated based on Eq. (9). As an on-line dynamical model, the topography changes in WRF-CHEM can lead to dynamical changes, such as the wind speeds at the foothill of the mountains. This is a useful and traditional sensitivity analysis method for numerical model to quantify the mountains effects, but with some shortcomings, which are to bring uncertainties to the sensitivity experiment. Firstly, the impact of topography is complicated to be completely quantified only by the altitude remove behavior. Secondly, the initial NCEP FNL data with mountains is treated as "real" in scenarios without mountains.**"**

b. The guiding effect is treated as part of the mountain blocking effect. We modified and added the description in the revised paper.

, "…through the blocking effect. The mountains block and redirect the airflows, causing the pollutant accumulations along the foothill of mountains. This study suggests that the prohibition of CFB should be strict not just in or around Beijing, but also on the ulterior crop growth areas of SNCP."

, "Here, it is attributed to the mountain blocking effect, which has two categories of influences. Firstly, the mountains block the airflows, causing pollutant accumulation and resulting in high $PM_{2.5}$ loading at the foothill of mountains (Influence-1, block). Secondly, the mountains redirect the airflows, causing the pollutants move toward the downwind foothill areas (Influence-2, redirect)."

, "Another major finding is that the mountains, surrounding the NCP in the north and west, play significant roles in enhancing the $PM_{2.5}$ pollution in NNCP through the blocking effect. Mountains block and redirect the airflows, causing the pollution accumulation along the foothill of mountains. The Taihang Mountains had greater impacts on $PM_{2.5}$ concentration than the Yanshan Mountains."

Supplementary data of Fig. S3, "**Fig. S3** The schematic pictures of mountains effect along with the topography of the NCP region. (a) Mountains block the airflows and cause pollutants accumulated at the foothill of mountains. (b) Mountains redirect the airflows, and cause pollutants move toward the downwind foothill areas (Influence-2, redirect)."

7. *The spin-up time is only 12 hours. I think the spin- up time is not long enough to get balanced. Also I wonder if any nudging method is used in this study (it should be explained in method part). If so, the contribution might be changed due to the nudging. Third, the domain is not large enough to ignore the impacts of "real" boundary condition. The mountainous topography may change the large-scale circulation.*

We rerun the model to extend the spin-up time to 3 days (), and updated related results (, Fig. 11…). As a regional model, the boundary effect cannot be avoided in the

WRF-CHEM. We agree with the referee that change model domain and use nudging method can change the model results. But we also need to consider the balance between large domain and cost of computation time. As a result, we have tested for different sizes for the model domains (900km x 900km). We think that the current domain 1200km x 1800km is reasonable and large enough for considering the lateral boundary effects. The important mountains (Taihang and Yanshan) have included in this domain.

**Response to Referee #2**

We thank the Referee for the careful reading of the manuscript and helpful comments. According to the suggestions of the referee, the comments have been carefully addressed, and the paper is carefully revised. We believe that the revised paper has been significantly improved after addressing the comments of the referee. We respond to each specific comment below. The original comments by the Referee are shown in bold italics. Our reply is shown in blue.

**General comments:**

*1. This paper conducts a numerical study to investigate the impact of crop field burning and topography on haze pollution in the North China Plain. This is an interesting study and potentially will be useful for air quality management in this region. However, there are several important points need to be appropriately addressed before it can be accepted for publishing in Atmospheric Chemistry and Physics.*

We thank the referee for the careful reading and the valuable comments that helped improving our paper.

**Detailed comments:**

*1. The impacts of biomass burning and topography on air quality in the North China Plain certainly are interesting topics for policy makers. However, because modeling results are sensitive from case to case, from policy perspectives, such kind of study should be conducted for a longer period. For the crop filed burning, this paper only conducted a case study for one week in October. According to Figure 2 of this paper and other previous studies in China, the most intensive period of biomass burning in the eastern China is June. It's a little bit strange that the authors selected a case in October. In fact, from the results presented in Figure 5, it is quite clear that in this case the crop field burning activities didn't play an important role on air quality in both NNCP and SNCP if it was compared with an overall*

*broad peak of anthropogenic pollution. For suck kind of non-typical case, I don't know whether it is meaningful to make many statistics to compare the relative contributions in several tables (e.g. Tables 3-6). With such a short-term case, I would like suggest conducting more in-depth analysis to understand specific scientific questions other than a calculation of numbers.*

We thank the referee for the thoughtful comment. This comment deals with several issues, and our responses are as follows:

a. We selected a case study in October rather than in June, because the air pollution was extremely high (with maximum concentrations larger than 300 μg/m$^3$) in NNCP, including Beijing, the capital city of China. It is important to understand the CFB contribution to the heavy air pollution in the incident.

b. We agree with the referee that in order to get a solid conclusion from policy perspectives, multiply case studies are needed. We clarify that the purpose of this study is to get some insights of how could CFB affect the air quality in NNCP and Beijing under heavy haze condition. However, in order to get quantitative analysis, more and longer studies are needed.

Line 461-463, "We get some insights of how could CFB affect the air quality in NNCP and Beijing under heavy haze condition, though more and longer studies are needed to get more representative conclusions."

c. The selected study period is under a heavy aerosol pollution period. The relative daily average contributions reach to a maximum of 34% and 32% in SNCP and NNCP, respectively. However, by considering the heavy pollution (> 300 μg/m$^3$) in this period, a little over 30% is prominent.

Line 351-355, "Indeed, the CFB pollution plume go through a long-range transport to NNCP can cause an obvious increase to PM$_{2.5}$ concentration, with the maximum daily average contribution of 32% (**Table 5**). Such a high transported contribution indicates that the CFB is not only one of the significant local pollution sources, but also a considerable regional pollution source."

d. According to the referee's suggestions, we added more statistical results in Table 6, including the regional average changes in mass (μg/m3) and percentage (%), and the lag-time of CFB pollution of NNCP from SNCP.

Line 370-388, "The pattern comparisons between simulated and observed near-surface PM$_{2.5}$ concentrations (*TPM$_{2.5}$*) perform well (**Fig. 9 Left Panels**). Meanwhile, the regional average CFB contributions are shown in **Table 6**, including mass concentration and related percentage as well as the related lag-time of NNCP corresponding to SNCP. At T1, massive local pollutants are emitted from CFB in SNCP and the CFB plume had not yet been largely transported to NNCP (see **$CPM_{2.5}$ of Fig. 9 T1**). The CFB contribution is high in SNCP with 72.6 μg m$^{-3}$, accounting for 71% of the total PM$_{2.5}$, whereas the CFB contribution is low with 8.1 μg m$^{-3}$ in NNCP, only accounting for 21%. At T2, high CFB contribution occurred in both SNCP and NNCP with 37 μg m$^{-3}$, suggesting that plenty of CFB pollutants emitted from SNCP and had been transported to NNCP (see **$CPM_{2.5}$ of Fig. 9 T2**). At T3, CFB contribution rapidly reduced in SNCP with 20.2 μg m$^{-3}$ (13%). It is worth to note that the high CFB contribution with 50.4 μg m$^{-3}$ (58%) is still remained in NNCP (see **$CPM_{2.5}$ of Fig. 9 T3**). At T4, the CFB contribution largely decreased in both SNCP and NNCP (no more than 6%) (see **$CPM_{2.5}$ of Fig. 9 T4**). The lag-time of NNCP to SNCP are 7-12 hours, and gradually increase from T1 to T4, implicating that the effect of CFB remains in longer time in NNCP than in SNCP. The highest PM$_{2.5}$ concentrations are along the foothill of the Taihang Mountains (**Left panels of Fig. 9**), which may be related to the mountain effects.**"**

*2. In the second part of this paper, the authors conducted an interesting numerical experiment by removing topography in specific regions in WRF-Chem model. However, such kind of treatment may cause some inconsistency in the initial conditions of meteorological parameters and the terrain data, which need more spin-up time because WRF uses a terrain-following vertical coordinate. However, according to the modeling description of this paper, the spin-up time for the WRF-Chem simulation is only 12 hours. Since the authors aim to give a quantitative understanding of the topographic effects, a longer spin-up time, for example several days, is needed. In addition, same as the Comment #1, as a case study for several days, the quantitative results here will have large uncertainty for policy makers. I would like suggest giving a more in-depth discussion by touching some scientific questions related to mountain, such as the impact of mountain-valley breezes on the accumulation of air pollutants etc.*

a. To address the comment of the referee, we extended the model spin-up time from 12 hours to 3 days (Line 178), and updated related results. We also added some shortcomings of the model study in the **Section 4.5 Impact of mountains**. As we stated in the above, we cannot give a quantitative analysis by the case study from policy perspectives, more cases and longer studies are needed.

Line 396-405, "In this study, we utilized the differences between the simulations with or without mountains to represent the effect of the topography on PM$_{2.5}$ concentration, which were calculated based on Eq. (9). As an on-line dynamical model, the topography changes in WRF-CHEM can lead to dynamical changes, such as the wind speeds at the foothill of the mountains. This is a useful and traditional sensitivity analysis method for numerical model to quantify the mountains effects, but with some shortcomings, which are to bring uncertainties to the sensitivity experiment. Firstly, the impact of topography is complicated to be completely quantified only by the altitude remove behavior. Secondly, the initial NCEP FNL data with mountains is treated as "real" in scenarios without mountains.**"**

b. The guiding effect is treated as part of the mountain blocking effect. We modified and added the description in the revised paper.

Line 31-34, "…through the blocking effect. The mountains block and redirect the airflows, causing the pollutant accumulations along the foothill of mountains. This study suggests that the prohibition of CFB should be strict not just in or around Beijing, but also on the ulterior crop growth areas of SNCP."

Line 418-423, "Here, it is attributed to the mountain blocking effect, which has two categories of influences. Firstly, the mountains block the airflows, causing pollutant accumulation and resulting in high $PM_{2.5}$ loading at the foothill of mountains (Influence-1, block). Secondly, the mountains redirect the airflows, causing the pollutants move toward the downwind foothill areas (Influence-2, redirect)."

Line 486-491, "Another major finding is that the mountains, surrounding the NCP in the north and west, play significant roles in enhancing the $PM_{2.5}$ pollution in NNCP through the blocking effect. Mountains block and redirect the airflows, causing the pollution accumulation along the foothill of mountains. The Taihang Mountains had greater impacts on $PM_{2.5}$ concentration than the Yanshan Mountains."

Supplementary data of Fig. S3, "**Fig. S3** The schematic pictures of mountains effect along with the topography of the NCP region. (a) Mountains block the airflows and cause pollutants accumulated at the foothill of mountains. (b) Mountains redirect the airflows, and cause pollutants move toward the downwind foothill areas (Influence-2, redirect)."

***3. The authors didn't give appropriate literature review for the both topics of biomass burning and topographic effect. In the model description part, the authors gave too many (more than 15) unnecessary references related to some common model schemes in WRF-Chem with some of them published several decades ago. However, in the main results part (Sect. 4), only two references (Cao et al., 2008 & Huang et al., 2012) are cited in the first paragraph but there is a lack of some comparisons of the results and conclusions with previous works done by other scientists for similar topics.***

To address the comment of the referee, we added several revisions.

a. A comprehensive summary of biomass burning emission topographic effect have been added in Line 52-66 and Line 80-91, respectively.

Line 52-66, "However, CFB have adverse impacts on traffic conditions and ecology environments (Shi et al., 2014;Zhang, 2009), and release plenty of pollutants, such as CO, $SO_2$, VOC, NOx and $PM_{2.5}$ (Koppmann et al., 2005;Li et al., 2008). According to Guan et al. (2014) and Lu et al. (2011), annual CFB contribute about 13% of the total particulate matter (PM) emissions in China (Zhang et al., 2016). And it is more prominent during the harvest periods due to its strong seasonal dependence. Numerous studies have quantified the contribution of biomass burning and CFB to PM pollution in China. According to Yao et al. (2016), Cheng et al. (2013), Wang et al. (2009; 2007) and Song et al. (2007), biomass burning has important impacts on the ambient $PM_{2.5}$ concentrations (15-24% in Beijing and 4-19% in Guangzhou). Yan et al. (2010) captured a heavy pollution with $PM_{10}$ concentrations higher than 350 $\mu g\ m^{-3}$ in some CFB locations. It is reported that CFB may contribute more than 30% of the $PM_{10}$ increase during CFB incidents (Zhu et al., 2012; Zha et al. 2013;Su et al., 2012). Cheng et al. (2014) report a summer case that CFB contributed 37% of $PM_{2.5}$ concentrations in the Yangtze River delta."

Line 80-91, "Yanshan and Taihang Mountains surround the NCP in the north and west (**Fig. 1c**). Such topography affects air pollution though PBL in complex ways (Miao et al., 2015b;Sun et al., 2013;Liu et al., 2009). Hu et al. (2014) have reported that the Loess Plateau and NCP result in a mountain-plains solenoid circulation, exacerbating air pollution over NCP. Chen et al. (2009) have founded that a mountain chimney effect is dominated by mountain-valley breeze, enhancing the surface air pollution in Beijing. The mountain-plain breeze develops frequently in Beijing and may play important roles in modulating the local air quality (Miao et al., 2015b;Hu et al., 2014;Chen et al., 2009). Miao et al. (2015a) founded that the mountains played a significant role in the sea-land aerosol circulation and the pollutants could be transported and accumulated in the NCP areas along the mountains, which is treated as the blocking effect (Zhao et al., 2015)."

b. A more detailed model description was added in **Section 3.1 Model description**.

Line 153-160, "The specific version of WRF-CHEM model is developed by Li et al. (2010; 2011; 2012), with a new flexible gas phase chemical module and the CMAQ (version 4.6) aerosol module developed by US EPA (Binkowski and Roselle, 2003). The wet deposition follows the CMAQ method and the dry deposition is parameterized following Wesely (1989). The photolysis rates are calculated using the FTUV (Li et al., 2005;Tie et al., 2003), in which the impacts of aerosols and clouds on the photochemistry are considered (Li et al., 2011)."

Line 162-166, "Meanwhile, the ISORROPIA Version 1.7 (http://nenes.eas.gatech.edu/ISORROPIA/) is utilized to simulate the inorganic aerosols, which is primarily used to predict the thermodynamic equilibrium between the ammonia-sulfate-nitrate-chloride-water aerosols and their gas phase precursors of $H_2SO_4$-$HNO_3$-$NH_3$-HCl-water vapor."

Line 184-185, "The biogenic emissions are calculated on-line with the WRF-CHEM model using the MEGAN model (Guenther, 2006)."

c. We modified and added explicit statements of provincial CFB emission inventory processing in Line 192-197 and Line 210-221. And we updated the provincial statistical data and related results. The detailed results and related references were added i n supplementary data of

Line 192-197, "This situation may be resulted from the limitation of local enforcement of regulation despite CFB have already been banned (Zhang and Cao, 2015;Shi et al., 2014). The CFB have a seasonal pattern due to the post-harvest activities with two distinct peaks in summer and autumn, especially in June (33-59%) and October (6-19%) (**Fig. 2b**). The strong seasonal dependence character suggests that the CFB emissions during October are much larger than annual averages."

Line 210-221, "where i stands for each province and k for different crop species of rice, corn and wheat. $E_{i,co}$ stands for CO emission from CFB of *i-th* province in gigagrams [Gg]. $P_{i,k}$ is the yield of crop in Gg. $F_i$ is the proportion of residues burned in the field. $D_k$ is the dry fraction of crop residue (dry matter). $R_k$ is the residue-to-crop ratio (dry matter). $CE_k$ is the combustion efficiency and $EF_{co}$ is the emission factors of CFB. The $P_{i,k}$ values were taken from an official statistical yearbook (NBS, 2015) (**Table S1**), and the $F_i$ on a provincial basis were taken from Wang and Zhang (2008) and Zhang Yisheng (Unpublished doctor thesis-in Chinese) (**Table S1**). The parameters of $D_k$, $R_k$, and $CE_k$ are listed in **Table S2**. The $EF_{co}$ from CFB was summarized range from 52 to 141 g kg$^{-1}$ in China (**Table S3**). In this study, we used 111 g kg$^{-1}$ as the average $EF_{co}$ of crop residue, which was used to estimate the emissions from global open burning (Wiedinmyer et al., 2011)."

**Impact of crop field burning and mountains on heavy haze in the North China Plain: A case study**

X. Long[1,2], X. X. Tie[1,3,4,*], J. J. Cao[1,5], R. J. Huang[1,6,*], T. Feng[1], N. Li[1,7], S. Y. Zhao[1], J. Tian[1], G. H. Li[1], Q. Zhang[8]

(1) Key Lab of Aerosol Chemistry & Physics, SKLLQG, Institute of Earth Environment, Chinese Academy of Sciences, Xi'an, 710061, China (2) University of Chinese Academy of Sciences, Beijing, 100049, China (3) CAS Center for Excellence in Urban Atmospheric Environment (CEUAE), Xiamen, 361021, China (4) National Center for Atmospheric Research, Boulder, CO, 80303, USA

(5) Institute of Global Environmental Change, Xi'an Jiaotong University, Xi'an, 710049, China (6) Laboratory of Atmospheric Chemistry, Paul Scherrer Institute (PSI), 5232 Villigen, Switzerland.

(7) Department of Atmospheric Science, National Taiwan University, Taipei, 10617, Taiwan (8) Center for Earth System Science, Tsinghua University, Beijing, 100084, China

*Correspondence to*: X. X. Tie (xxtie@urcar.edu) R. J. Huang (rujin.huang@ieecas.cn)

**Abstract.** With the provincial statistical data and CFB activities captured by MODIS, we extracted a detailed CFB emission inventory in the North China Plain (NCP). The WRF-CHEM model is applied to investigate the impact of CFB on air pollution during the period from October 6 to 12, 2014, corresponding to a heavy haze incident with high concentrations of $PM_{2.5}$ (particulate matter with aerodynamic diameter less than 2.5 μm). The WRF-CHEM model generally performs well in simulating the surface species concentrations of $PM_{2.5}$, $O_3$ and $NO_2$ compared to the observations. And reasonably reproduced the observed temporal variations of wind speed, wind direction and planetary boundary layer height (PBLH). It is found that the CFB occurred in southern NCP (SNCP) have significant effects on $PM_{2.5}$ concentrations locally, causing a maximum of 34% $PM_{2.5}$ increase. Under the continuously southerly wind condition, the CFB pollution plume go through a long-range transport to northern NCP (NNCP-with several mega cities, including Beijing, the capital city of China), where few CFB occurred, resulting in a maximum of 32% $PM_{2.5}$ increase. As a result, the heavy haze in Beijing is enhanced by the CFB occurred in SNCP. Mountains also play significant roles in enhancing the $PM_{2.5}$ pollution in NNCP through the blocking effect. The mountains block and redirect the airflows, causing the pollutant accumulations along the foothill of mountains. This study suggests that the prohibition of CFB should be strict not just in or around Beijing, but also on the ulterior crop growth areas of SNCP. $PM_{2.5}$ emissions in SNCP should be significantly limited in order to reduce the occurrences of heavy haze events in NNCP region.

**Key words:** crop field burning; mountain effect; $PM_{2.5}$; WRF-CHEM

**1 Introduction**

Crop residue burning is important for global biomass burning (Yevich and Logan, 2003;Shon, 2015), especially in agricultural countries such as China. Crop residue resources in China rank the first in the world, accounting for 17.3% of the global production (Bi et al., 2010), and increasing with the average annual proportion of 4% (Hong et al., 2015;Zhao et al., 2010). Compared with other approaches, crop field burning (CFB) is the most effective and less expensive to remove residues. The national annual average proportion of CFB to total residues is about 11-25%(Cao et al., 2008;Hao and Liu, 1994;Streets et al., 2003;Wang and Zhang, 2008;Zhao et al., 2010). Large numbers of annual CFB occur in China (Zhang et al., 2015; Yan et al., 2006), especially during the post-harvest seasons (Zhang et al., 2016;Shi et al., 2014;Cao et al., 2008). And most of the CFB occur on crop growth areas, such as the North China Plain (NCP) (Huang et al., 2012;Li et al., 2008), where have been frequently suffering haze events in recent years (Yang et al., 2015;Jiang et al., 2015;Wang et al., 2013;Wang et al., 2012).

However, CFB have adverse impacts on traffic conditions and ecology environments (Shi et al., 2014;Zhang, 2009), and release plenty of pollutants, such as CO, $SO_2$, VOC, NOx and $PM_{2.5}$ (Koppmann et al., 2005;Li et al., 2008). According to Guan et al. (2014) and Lu et al. (2011), annual CFB contribute about 13% of the total particulate matter (PM) emissions in China (Zhang et al., 2016). And it is more prominent during the harvest periods due to its strong seasonal dependence. Numerous studies have quantified the contribution of biomass burning and CFB to

PM pollution in China. According to Yao et al. (2016), Cheng et al. (2013), Wang et al. (2009; 2007) and Song et al. (2007), biomass burning has important impacts on the ambient $PM_{2.5}$ concentrations (15-24% in Beijing and 4-19% in Guangzhou). Yan et al. (2010) captured a heavy pollution with $PM_{10}$ concentrations higher than 350 μg $m^{-3}$ in some CFB locations. It is reported that CFB may contribute more than 30% of the $PM_{10}$ increase during CFB incidents (Zhu et al., 2012; Zha et al. 2013;Su et al., 2012). Cheng et al. (2014) report a summer case that CFB contributed 37% of $PM_{2.5}$ concentrations in the Yangtze River delta.

The impact of CFB to air quality is continental and regional. Air quality in China is influenced by the CFB occurred in Southeast Asia and on the Indian Peninsula (Qin et al., 2006). Mukai et al. (2014) have reported that CFB emissions in Southeast Asia contribute the carbonaceous aerosols in Beijing. Within China, the inter-province transported air pollutants emitted from CFB significantly affect regional PM levels and air quality (Cheng et al., 2014;Zhu et al., 2012). For Beijing, the smoke particles from CFB are expected to be one of the major components (Wang et al., 2014;Cheng et al., 2013), though the percentage of transported sources are seldom specified (Zhang et al., 2016). A recent study reports that CFB and regional transport illustrate two of the key processes of haze formation in October 2014, especially on Oct. 6[th], but without quantitative estimation in this work (Yang et al., 2015). Related quantification studies are of great importance for the control strategies of CFB in Beijing.

Yanshan and Taihang Mountains surround the NCP in the north and west (**Fig. 1c**). Such topography affects air pollution though PBL in complex ways (Miao et al., 2015b;Sun et al., 2013;Liu et al., 2009). Hu et al. (2014) have reported that the Loess Plateau and NCP result in a mountain-plains solenoid circulation, exacerbating air pollution over NCP. Chen et al. (2009) have founded that a mountain chimney effect is dominated by mountain-valley breeze, enhancing the surface air pollution in Beijing. The mountain-plain breeze develops frequently in Beijing and may play important roles in modulating the local air quality (Miao et al., 2015b;Hu et al., 2014;Chen et al., 2009). Miao et al. (2015a) founded that the mountains played a significant role in the sea-land aerosol circulation and the pollutants could be transported and accumulated in the NCP areas along the mountains, which is treated as the blocking effect (Zhao et al., 2015).

In this study, we analyzed a heavy haze episode occurred in NCP region from "LT" 12:00 6$^{th}$ to 00:00 12$^{th}$ October in 2014. During the period, the average PM$_{2.5}$ concentrations are much higher than class II standard in both SNCP and NNCP. The characteristics of the air pollution were analyzed based on PM$_{2.5}$ concentration. Depending on the satellite-based observations of Moderate Resolution Imaging Spectroradiometer (MODIS), a large number of CFB occurred in SNCP, whereas few CFB happened in NNCP. A more detailed CFB emission inventory was extracted. Thereafter we analyzed the regional transport of CFB emissions from SNCP to NNCP driven by prevailing southerly winds. Under the continuously southerly wind condition, the mountains play important roles for the northward transport, and cause the accumulation of the aerosol pollutants at the foothill of the mountains. We also analyzed the impact of mountains (especially the Taihang Mountains and the Yanshan Mountains) on the air pollution transport.

**2 Description of data**

**2.1 Geographical Location**

In order to study the effect of CFB on local and regional air pollution, the research domain locates in eastern China, covering a large regional area (more than 10 provinces) (see **Fig. 1a**). The NCP region is in the middle of the research domain, with two mountains in the north and west. The Yanshan Mountains locate in the north of NCP with east-west directions, and the Taihang Mountains locate in the west of NCP with southwest-northeast directions (**Fig. 1b**). **Figure 1c** displays the distribution of online sampling sites and CFB captured by MODIS during the haze episodes. We defined two regions of interests according to CFB occurrences, topographic conditions, industrial and agricultural developments. One is the northern NCP (NNCP), including two mega cities (Beijing and Tianjin) and the north part of Hebei province, where only few CFB occurred. Another is the southern NCP (SNCP), where substantial CFB occurred during the haze episodes as shown in **Fig. 1c**. Because of the severe haze problem in the capital city of China (Beijing), one of the main focuses is to study the long-range transport of CFB pollution from SNCP to NNCP.

**2.2 Meteorological conditions**

The reanalysis meteorological data, including wind direction, wind speed and PBLH were obtained from the European Centre for Medium-range Weather Forecasts (ECMWF), with a spatial resolution of $0.125° \times 0.125°$. The data is available at: http://www.ecmwf.int/products/data/. The average wind direction and wind speed are displayed in **Table 1**. During the haze episode, the mean wind directions are $174.8°$ in NNCP and $165.2°$ in SNCP, and the average wind speeds are $2.4$ m s$^{-1}$ in both NNCP and SNCP. The results suggest that the prevailing winds are continuously southerly winds with weak wind speeds, which is favorable to the pollution long-range transport from SNCP to NNCP, which has been reported as one of the major contributors to haze formation in the Beijing City (Tie et al., 2015).

**2.3 PM$_{2.5}$ Measurements**

The hourly PM$_{2.5}$ mass concentration were continuously monitored by the Ministry of Environmental Protection (MEP) of China (http://datacenter.mep.gov.cn), including 5 sites in NNCP and 7 sites in SNCP (indicated by green crosses in **Fig. 1c**). The data was updated from the website: http://www.pm25.in/. **Table 1** summarizes the site information and the observed PM$_{2.5}$ concentrations. During the study period, the average PM$_{2.5}$ concentrations are $200.0$ μg m$^{-3}$ in NNCP and $184.1$ μg m$^{-3}$ in SNCP. The measured PM$_{2.5}$ concentrations are much higher than class II standard (daily mean of $75$ μg m$^{-3}$), indicating an occurrence of heavy pollution event. We analyzed the characteristics of the air pollution based on PM$_{2.5}$ concentration simulated by

WRF-CHEM. Meanwhile, it is worth to note that the highest $PM_{2.5}$ concentrations occurred along the foothill sites of the Taihang Mountains. At the foothill sites of BJ,

BD, SJZ and XT, $PM_{2.5}$ concentrations are 245.5, 287.7, 257.9, and 320.1 μg m$^{-3}$, respectively. The mean $PM_{2.5}$ concentration in these 4 sites is 277.8 μg m$^{-3}$, much higher than 147.2 μg m$^{-3}$ averaged from the other sites. Considering the continuously southerly winds and the topographic conditions, we studied the impact of the mountains on the air pollution transport.

**3 Methods**

**3.1 Model description**

We use Weather Research and Forecasting Chemical model (WRF-CHEM) (Grell et al., 2005) to simulate the spatial and temporal variability of $PM_{2.5}$ concentration. The specific version of WRF-CHEM model is developed by Li et al. (2010; 2011; 2012), with a new flexible gas phase chemical module and the CMAQ (version 4.6) aerosol module developed by US EPA (Binkowski and Roselle, 2003). The wet deposition follows the CMAQ method and the dry deposition is parameterized following Wesely (1989). The photolysis rates are calculated using the FTUV (Li et al., 2005;Tie et al.,

2003), in which the impacts of aerosols and clouds on the photochemistry are considered (Li et al., 2011). The gas-phase chemistry was represented in the model by the modified RADM2 (Regional Acid Deposition Model, version 2) gas-phase chemical mechanism (Stockwell et al., 1990;Chang et al., 1987). Meanwhile, the

ISORROPIA Version 1.7 (http://nenes.eas.gatech.edu/ISORROPIA/) is utilized to simulate the inorganic aerosols, which is primarily used to predict the thermodynamic equilibrium between the ammonia-sulfate-nitrate-chloride-water aerosols and their gas phase precursors of $H_2SO_4$-$HNO_3$-$NH_3$-HCl-water vapor. The Yonsei University (YSU) PBL scheme (Hong et al., 2006), Lin microphysics scheme (Lin et al., 1983),

Noah land-surface model (Chen and Dudhia, 2001) were utilized. The model has been successfully applied in several regional pollution studies (Tie et al., 2009;Tie et al.,

2007;He et al., 2015).

The WRF-CHEM model is configured with resolution of $6 \times 6$ km ($200 \times 300$ grid cells) centered in (117°E, 39°N). Vertical layers extend from the surface to 50 hPa, with 28 vertical layers, involving 7 layers in the bottom of 1 km. The meteorological initial and boundary conditions were gathered from NCEP FNL Operational Global

Analysis data. The lateral chemical initial conditions were constrained by a global chemical transport model-MOZART4 (Model for Ozone and Related chemical

Tracers, Version 4) 6-hour output (Emmons et al., 2010;Tie et al., 2005). For the episode simulations, the spin-up time of the WRF-CHEM model is 3 days.

The surface emission inventory used in this study was obtained from the

Multi-resolution Emission Inventory for China (MEIC) (Zhang et al., 2009), which is an update and improvement for the year 2010 (http://www.meicmodel.org). The emission inventory estimated only anthropogenic emission such as non-residential sources (transportation, agriculture, industry and power) and residential sources related to fuel combustions. The biogenic emissions are calculated on-line with the

WRF-CHEM model using the MEGAN model (Guenther, 2006). Additionally, we added emission from CFB in the present study.

**3.2 Crop field burning emissions**

We analyzed the annual and monthly number of open crop fire events captured by

MODIS in the research domain from 2008 to 2014. In the NCP region, the CFB

activities gradually increase from the minimum fire events of 12, 000 times in 2008 to

27, 000 times in 2014 (**Fig. 2a**), suggesting that the CFB is not efficiently controlled in this region. This situation may be resulted from the limitation of local enforcement of regulation despite CFB have already been banned (Zhang and Cao, 2015;Shi et al.,

2014). The CFB have a seasonal pattern due to the post-harvest activities with two distinct peaks in summer and autumn, especially in June (33-59%) and October (6-19%) (**Fig. 2b**). The strong seasonal dependence character suggests that the CFB

emissions during October are much larger than annual averages. In order to have the detailed horizontal distribution of the pollutant emissions of CFB, we elaborated a method to generate emission inventory using the satellite data of "MODIS Thermal

Anomalies/Fire product (MOD/MYD14DL)". The MOD/MYD14DL product can detect small opening fires ($<100 \text{ m}^2$) (Giglio et al., 2003) and produce the geographic location of open fire activities (van der Werf et al., 2006). In this study, the CFB was defined as MOD/MYD14DL active fires occurred over the cropland, which is classified by the MODIS Combined Land Cover Type product (Friedl et al., 2010).

Firstly, we estimated the CO emission of CFB, utilizing a widely used method (Streets et al., 2003;Cao et al., 2008;Zhang et al., 2008;Ni et al., 2015a) based on the annual provincial statistical data. The provincial emission of crop residues burning can be calculated by Eq. (1):

$E_{i,CO} = \sum_{k=1}^{3} P_{i,k} \times F_i \times D_k \times R_k \times CE_k \times EF_{co}$          (1)

where i stands for each province and k for different crop species of rice, corn and wheat. $E_{i,co}$ stands for CO emission from CFB of *i-th* province in gigagrams [Gg]. $P_{i,k}$

is the yield of crop in Gg. $F_i$ is the proportion of residues burned in the field. $D_k$ is the dry fraction of crop residue (dry matter). $R_k$ is the residue-to-crop ratio (dry matter). $CE_k$ is the combustion efficiency and $EF_{co}$ is the emission factors of CFB.

The $P_{i,k}$ values were taken from an official statistical yearbook (NBS, 2015) (**Table**

**S1**), and the $F_i$ on a provincial basis were taken from Wang and Zhang (2008) and

Zhang Yisheng (Unpublished doctor thesis-in Chinese) (**Table S1**). The parameters of

$D_k$, $R_k$, and $CE_k$ are listed in **Table S2**. The $EF_{co}$ from CFB was summarized range from 52 to 141 g kg$^{-1}$ in China (**Table S3**). In this study, we used 111 g kg$^{-1}$ as the average $EF_{co}$ of crop residue, which was used to estimate the emissions from global open burning (Wiedinmyer et al., 2011).

The provincial CO emission was temporally and spatially allocated according to the

CFB activities. The detailed daily CO emission of *k-th* grid ($E_{k,co}$) was calculated using Eq. (2):

$E_{k,CO} = \frac{FC_k}{FC_i} \times E_{i,CO}$ ,          (2)

where $FC_k$ and $FC_i$ are the total CFB fire counts in *k-th* grid and *i-th* province, respectively (**Table S1**).

Thereafter, the emissions of various gaseous and particulate species ($E_{spec1}$) were calculated by the Eq. (3). And individual chemical compounds ($E_{spec2}$) were calculated by Eq. (4).

$$E_{k,spec1} = \frac{EF_{spec1}}{EF_{CO}} \times E_{k,CO} \text{ ,} \tag{3}$$

$$E_{k,spec2} = E_{k,NMOC} \times \text{scale,} \tag{4}$$

where $E_{k,spec1}$ and $E_{k,spec2}$ are the *k-th* grid emission of the specify WRF-CHEM

species; $E_{spec1}$ and $EF_{CO}$ are the emission factors of CFB; $E_{k,NMOC}$ is NMOC emission in the *k-th* grid calculated by Eq. (3); *scale* is the value to convert NMOC emissions to WRF-CHEM chemical species. The emission factors for gaseous and particulate species and *scale* to convert NMOC emissions to WRF-CHEM chemical species from

CFB were taken from available datasets (Wiedinmyer et al., 2011;Akagi et al.,

2011;Andreae and Merlet, 2001), which were summarized by Wiedinmyer et al.

(2011) (**Table 2**).

**4 Results and discussions**

**4.1 Evaluate the Crop field burning emission**

The provincial CO emissions of CFB were estimated based on Eq. (1), and there was

8.2 Tg CO emitted from CFB in 2014 (**Table S1**). This result is comparable to previous studies, which is 4.6-10.1 Tg yr[-1] (Cao et al., 2008;Ni et al., 2015;Streets et al., 2003;Yan et al., 2006). According to the MODIS observations, a large number of

CFB occurred in SNCP, including provinces of Henan with 61% and Shandong with

22%. Most of CFB occurred on Oct. 6[th] and 7[th], accounting for 75% (**Table 3**).

**Table 4** shows the CFB emissions of gaseous and particulate species on Oct. 6[th] and

7[th], including the mega cities of Beijing and Tianjin, and provinces of Hebei, Henan and Shandong in NCP. **Figure 3** displays the CFB activities and related CO emission on Oct. 6[th] and 7[th]. Most of the pollutants are emitted from Henan in SNCP, accounting for 73% on Oct. 6[th] and 65% on Oct. 7[th]. Plenty of pollutants emitted from

CFB on Oct. 6[th], producing more than 5.1 Gg PM$_{2.5}$ and 98.0 Gg CO (1 Gg = 10$^9$ g).

**4.2 Statistical characteristics of the evaluation**

The characteristics of the haze pollution were defined by PM$_{2.5}$ concentration, which is significantly affected by the local wind fields and PBLH in the NCP region (Tie et al., 2015). In order to evaluate the model performance, the model simulations were compared with the measured results in both species concentrations (PM$_{2.5}$, O$_3$ and

NO$_2$) and meteorological parameters (wind speed, wind direction and PBLH). The normalized mean bias (NMB) and correlation coefficient (R) were used to quantify the performance.

$$NMB = \frac{\sum_{i=1}^{N}(P_i - O_i)}{\sum_{i=1}^{N} O_i},$$ (5)

$$R = \frac{\sum_{i=1}^{N}(P_i - \bar{P})(O_i - \bar{O})}{[\sum_{i=1}^{N}(P_i - \bar{P})^2 \sum_{i=1}^{N}(O_i - \bar{O})^2]^{\frac{1}{2}}},$$ (6)

where $P_i$ is the predicted results and $O_i$ represents the related observations. N is the total number of the predictions used for comparisons. Meanwhile, $\bar{P}$ and $\bar{O}$ are the average prediction and related mean observation, respectively.

**Figure 4** shows the measured and calculated temporal variations of regional average species concentrations, including PM$_{2.5}$, O$_3$ and NO$_2$. The WRF-CHEM model reproduced the pollution episode well, with a good agreement with observations. The correlation coefficients (R) of simulated and measured $PM_{2.5}$ concentrations are 0.88

in both NNCP and SNCP (**Fig. 4a**). The simulations are overall lower than the observations with NMB of -12% in NNCP and -7% in SNCP. Considering the high average $PM_{2.5}$ concentration with 200.0 $\mu g\ m^{-3}$ in NNCP and 184.1 $\mu g\ m^{-3}$ in SNCP, obvious underestimates exist with the overall concentrations of 24.0 $\mu g\ m^{-3}$ in NNCP

and 12.9 $\mu g\ m^{-3}$ in SNPC. This may be related to the CMAQ (version 4.6) aerosol module, which is likely to underestimated OM due to the uncertainty in secondary organic aerosols mechanism (Baek et al., 2011). Meanwhile, the underestimates are also related to the negative bias in S3, which may be related to cloud contamination (**Fig. S1**). Whereas this has only a few impacts on the estimation of CFB contribution since few CFB occurred during S3. The simulations of $O_3$ and $NO_2$ are also agree well with observations, with R greater than 0.77 and absolute NMB lower than 17% (**Fig.**

**4b and 4c**). **Figure 5** displays the measured and calculated temporal variations of regional average meteorological parameters, including wind speed, wind direction, and the PBLH in both NNCP and SNCP. The comparisons between simulated and observed wind fields show good agreements (**Fig. 5a and 5b**), with all the R higher than 0.64, and the absolute NMB are no more than 15%. Meanwhile, the R of PBLH

is larger than 0.88 and the absolute NMB is no more than 10% (**Fig. 5c**).

**4.3 Characteristics of the heavy pollution events**

According to the evolution of $PM_{2.5}$ concentration (**Fig. 4a**), the haze episode can be divided into three stages: (I) pollution formation stage (S1, 12:00 6$^{th}$ - 00:00 8$^{th}$), (II) pollution outbreak stage (S2, 00:00 8$^{th}$ - 00:00 10$^{th}$) and (III) pollution clear stage (S3, 00:00 10$^{th}$ - 00:00 12$^{th}$). The major characteristics of each stage are briefly summarized below. Related simulations in bracket follow the detailed observations.

- S1 (pollution formation): It is dominated by a continuously southerly wind, with mean wind speed of 2.5 (2.7) m s$^{-1}$ in NNCP and 3.0 (3.6) m s$^{-1}$ in SNCP. The backward trajectories, with the HYSPLIT model online version, of BJ, TJ and BD during S1 reflected how the CFB influenced the NNCP region (**Fig. 6**). The air mass mainly came from the south, originating from the SNCP region. The pollutants are continuously transported from SNCP to NNCP, leading to pollutants accumulation in NNCP, which is characterized by the steady rising of PM$_{2.5}$ concentration in NNCP from 20.6 (41.0) μg m$^{-3}$ (at 12:00 Oct. 6$^{th}$) to 242.7 (217.5) μg m$^{-3}$ (at 00:00 Oct. 8$^{th}$) (**Fig. 4 a1**).

- S2 (pollution outbreak): During S2, the air pollution deteriorates. It is a relative stable period of heavy pollution with average PM$_{2.5}$ concentration of 252.0 (241.2) μg m$^{-3}$ in NNCP and 214.1 (235.0) μg m$^{-3}$ in SNCP, which are higher than those in other stages. This phenomenon may be related to the relative lower wind speed and PBLH.

- S3 (pollution clear): During S3, the southerly winds gradually decrease, and turn to be northerly at the end of S3. Clean airs from the north region of China obviously improve the air quality. Compared with S2, the average PM$_{2.5}$ concentrations are decreased in both NNCP and SNCP.

There were several important issues shown in the results, and should be addressed. (1)

The PM$_{2.5}$ concentrations are extremely high during the S2 period, and the daily average concentrations are greater than the Chinese National Standard (75 μg m$^{-3}$) by

2-3 times. (2) The air pollutions are severe in a large region (occurred in both NNCP

and SNCP). (3) During the S1 and S2 periods, there is a time lag between SNCP and

NNCP for PM$_{2.5}$ concentrations. Because it is a continuously southerly wind condition, it shows the important impact of long-range transport of PM$_{2.5}$ particles from the

SNCP to NNCP.

**4.4 Contributions of crop field burning**

Model sensitivity studies were conducted to separate the individual CFB contribution.

Two model simulations were performed, i.e., one with both anthropogenic and CFB

emissions while the other with only anthropogenic emission. We calculated PM$_{2.5}$

distributions by including CFB emissions (anthropologic and CFB) and excluding

CFB emissions (only anthropologic). In this study, the CFB contributions were quantified by regional average contribution in mass concentration ($CPM_{2.5}$) and daily average contribution proportion ($\overline{PPM_{2.5}}$).

$CPM_{2.5} = TPM_{2.5} - APM_{2.5},$ \hspace{3cm} (7)

$\overline{PPM_{2.5}} = \dfrac{\overline{CPM_{2.5}}}{\overline{TPM_{2.5}}},$ \hspace{3cm} (8)

where $TPM_{2.5}$ represents the simulated PM$_{2.5}$ concentrations considering total emission; $APM_{2.5}$ denotes the simulated PM$_{2.5}$ concentrations only considering anthropologic emissions. $\overline{CPM_{2.5}}$ and $\overline{TPM_{2.5}}$ are daily average value for $CPM_{2.5}$

and $TPM_{2.5}$, respectively.

**Figure 7** displays the regional observed and simulated $PM_{2.5}$ concentrations considering total emissions (anthropologic and CFB) and only anthropologic emissions. It is clearly shown that the CFB had important contributions to $PM_{2.5}$ in both NNCP (**Fig. 7a)** and SNCP **Fig. 7b**). This is also proved by the daily average contribution proportion ($\overline{PPM_{2.5}}$) of CFB (**Table 5**). The high values of $\overline{PPM_{2.5}}$ in

SNCP appear on Oct 6$^{th}$ with 34% and on 7$^{th}$ with 17%, when plenty of CFB occurred.

Simultaneously, the high values of $\overline{PPM_{2.5}}$ in NNCP appear on Oct 7$^{th}$ with 32% and

8$^{th}$ with 10%, showing a later occurrence than that in SNCP. The time lag suggests that the plume with CFB may be transported from SNCP to NNCP.

The detailed hourly CFB contributions to $PM_{2.5}$ concentrations ($CPM_{2.5}$) are displayed in **Fig. 8**. The values of $CPM_{2.5}$ in NNCP are generally lag synchronized with that in SNCP, such as $P_{N1}$ versus $P_{S1}$ and $P_{N2}$ versus to $P_{S2}$ (**Fig. 8a and 8b**).

Apparently, the lagged time is not constant and varied with the wind fields. The specific details perform relaxed lag synchronized, especially between the $P_{N2}$ and $P_{S2}$.

This phenomenon further indicates that the CFB contribution in SNCP is mainly due to local emission, whereas CFB contribution in NNCP is largely resulted from long-range transport from SNCP. Indeed, the CFB pollution plume go through a long-range transport to NNCP can cause an obvious increase to $PM_{2.5}$ concentration, with the maximum daily average contribution of 32% (**Table 5**). Such a high transported contribution indicates that the CFB is not only one of the significant local pollution sources, but also a considerable regional pollution source.

To clearly show the time evolution of the CFB effect on $PM_{2.5}$ concentration, four time-points were defined in **Fig. 8c**, such as T1 (23:00 6[th]), T2 (05:00 7[th]), T3 (20:00

7[th]) and T4 (19:00 8[th]). At T1, prominent CFB contribution occurred in SNCP with the highest value of 71.9 μg m$^{-3}$, but accompanied with unimportant CFB contribution in NNCP with a low value of 7.7 μg m$^{-3}$. At T2, the CFB contribution in SNCP

decline with a relative high value of 44.2 μg m$^{-3}$, but rise in NNCP with 51.6 μg m$^{-3}$

(near the transition between P1 and P2). At T3, the CFB contribution rapidly decreases to a low value of 24.0 μg m$^{-3}$ in SNCP, but increase to the highest with 47.0

μg m$^{-3}$ in NNCP. At T4, the CFB contributions largely decrease, becoming lesser in both SCNP (9.1 μg m$^{-3}$) and NNCP (11.4 μg m$^{-3}$). Interestingly, the CFB contribution in SNCP drops faster than that in NNCP (**P2 in Fig. 8c**), resulting in stronger effects in NNCP than in SNCP, as well as longer effects in NNCP.

To further understand the evolution of CFB to heavy haze pollution, we analyzed the horizontal distributions of $PM_{2.5}$ concentration ($TPM_{2.5}$) and related CFB contribution ($CPM_{2.5}$) at T1, T2, T3 and T4 (**Fig. 9**). The pattern comparisons between simulated and observed near-surface $PM_{2.5}$ concentrations ($TPM_{2.5}$) perform well (**Fig. 9 Left**

**Panels**). Meanwhile, the regional average CFB contributions are shown in **Table 6**, including mass concentration and related percentage as well as the related lag-time of

NNCP corresponding to SNCP. At T1, massive local pollutants are emitted from CFB

in SNCP and the CFB plume had not yet been largely transported to NNCP (see

*$CPM_{2.5}$ of Fig. 9 T1*). The CFB contribution is high in SNCP with 72.6 μg m$^{-3}$, accounting for 71% of the total $PM_{2.5}$, whereas the CFB contribution is low with 8.1

μg m$^{-3}$ in NNCP, only accounting for 21%. At T2, high CFB contribution occurred in both SNCP and NNCP with 37 μg m$^{-3}$, suggesting that plenty of CFB pollutants emitted from SNCP and had been transported to NNCP (see ***CPM$_{2.5}$* of Fig. 9 T2**). At

T3, CFB contribution rapidly reduced in SNCP with 20.2 μg m$^{-3}$ (13%). It is worth to note that the high CFB contribution with 50.4 μg m$^{-3}$ (58%) is still remained in NNCP

(see ***CPM$_{2.5}$* of Fig. 9 T3**). At T4, the CFB contribution largely decreased in both

SNCP and NNCP (no more than 6%) (see ***CPM$_{2.5}$* of Fig. 9 T4**). The lag-time of

NNCP to SNCP are 7-12 hours, and gradually increase from T1 to T4, implicating that the effect of CFB remains in longer time in NNCP than in SNCP. The highest

PM$_{2.5}$ concentrations are along the foothill of the Taihang Mountains (**Left panels of**

**Fig. 9**), which may be related to the mountain effects.

**4.5 Impact of mountains**

Sensitivity experiments were conducted to quantify the impacts of the Taihang

Mountains (referred as R-T), the Yanshan Mountains (R-Y) and both of them (R-TY)

on the heavy pollution. The mountains were removed from the model calculation, in which, the altitude of mountains were reduced to the average altitude of NCP (30 m).

With the reduction of altitudes of the topography, the dynamical conditions calculated from WRF-CHEM changed, which affect pollutions transport, especially along the foothill of mountains. In this study, we utilized the differences between the simulations with or without mountains to represent the effect of the topography on

PM$_{2.5}$ concentration, which were calculated based on Eq. (9). As an on-line dynamical model, the topography changes in WRF-CHEM can lead to dynamical changes, such as the wind speeds at the foothill of the mountains. This is a useful and traditional sensitivity analysis method for numerical model to quantify the mountains effects, but with some shortcomings, which are to bring uncertainties to the sensitivity experiment.

Firstly, the impact of topography is complicated to be completely quantified only by the altitude remove behavior. Secondly, the initial NCEP FNL data with mountains is treated as "real" in scenarios without mountains. The sensitive configuration and related enclosing scope are displayed in **Fig. S2.**

$$IPM_{2.5} = RPM_{2.5} - TPM_{2.5}, \tag{9}$$

where $IPM_{2.5}$ is the net impacts of mountains on $PM_{2.5}$; $RPM_{2.5}$ denotes the simulated

$PM_{2.5}$ concentration with removal behaviors, involving R-TY, R-T, and R-Y; $TPM_{2.5}$

represents the simulated $PM_{2.5}$ concentration considering emission of anthropologic and CFB, which is correspond with the case of R0 (**Fig. S2a**).

The sensitivity study period was selected from 12:00 7$^{th}$ to 00:00 10$^{th}$. **Fig. 10**

displays the elevation contours and the horizontal distributions of $PM_{2.5}$ concentration with the effect of mountains, exhibiting a good performance of the pattern comparisons between simulated and observed near-surface $PM_{2.5}$ concentrations. The results illustrate that the mountains had important impacts on regional $PM_{2.5}$

concentration, especially for the region along the foothill of mountains with a heavy pollution area, covering sampling sites of BJ, BD, SJZ and XT. Here, it is attributed to the mountain blocking effect, which has two categories of influences. Firstly, the mountains block the airflows, causing pollutant accumulation and resulting in high

PM$_{2.5}$ loading at the foothill of mountains (Influence-1, block). Secondly, the mountains redirect the airflows, causing the pollutants move toward the downwind foothill areas (Influence-2, redirect). Both influences act to prevent the pollutant plume to disperse toward western mountains, causing accumulations of the air pollutants along the foothill of mountains. These two influences of mountain blocking effect are illustrated as the schematic pictures in **Fig. S3**.

**Fig. 11** displays the simulated PM$_{2.5}$ concentration due to the mountain effects (*RPM$_{2.5}$*), with the three cases (R-TY, R-T, and R-Y). The heavy pollution accumulation **(Fig. 10)** along the foothill of mountains is significantly reduced, especially with the removal of Taihang Mountains (R-T, and R-TY) (**Fig. 11 a1** and

**a2**). In these two cases, the pollution plumes dispersed westerly (**Fig. 11 b1** and **b2**).

The PM$_{2.5}$ concentrations increase 40-120 μg m$^{-3}$ in the western part of Taihang

Mountains, and reduce 20-60 μg m$^{-3}$ in NCP. The distribution of the reduced pollution plume shows a northeast band plume, indicating the mountain blocking effect. With the removal of the Yanshan Mountains (R-Y), the high PM$_{2.5}$ concentrations are still remained along the foothill of the Taihang Mountains (**Fig. 11 a3**), but more pollutants are pushed forward along the foothill, toward the northeastern NCP.

Without the blocking effect of the Yanshan Mountains, the PM$_{2.5}$ concentrations increased 20-80 μg m$^{-3}$ in the northern part of the Yanshan Mountains, and decreased

10-60 μg m$^{-3}$ in the southern part of the Yanshan Mountains (**Fig. 11 b3**).

In the foothill sampling sites (BJ, BD, SJZ and XT), the average PM$_{2.5}$ concentrations are reduced 54.2 μg m$^{-3}$ for the case of R-T, which is much higher than the case of

R-Y ($28.4$ μg m$^{-3}$). For the other non-foothill sites, the average reduction is $34.7$ μg m$^{-3}$ for the case of R-T, which is also much higher than the case of R-Y ($2.4$ μg m$^{-3}$), suggesting that the Taihang Mountains have stronger effects than the Yanshan

Mountains. Meanwhile, the higher impacts in the foothill sampling sites than non-foothill sites are further demonstrated.

**5 Conclusions**

In recent years, the NCP region, including the capital city of Beijing, has been suffering serious haze pollution problem, especially in winter and summer. Most studies concerned about the intense secondary formation, huge regional transport of pollutants, stationary meteorological conditions and large local emission. In autumn,

CFB and movement of wind based on large scale topography are important in NCP, whereas the percentage of transported CFB emission sources are seldom specified.

This is probably resulted from the contingency of CFB activities during harvest period and the limitation of temporal resolution of CFB emission inventories. In this study, we extracted a more detailed CFB emission inventory based on the provincial statistical data and CFB activities captured by MODIS. The WRF-CHEM mode was applied to study the effect of CFB on the PM$_{2.5}$ concentrations in NCP, especially the evaluation of CFB plums pollution, such as local influence and long-range transportation. We get some insights of how could CFB affect the air quality in NNCP

and Beijing under heavy haze condition, though more and longer studies are needed to get more representative conclusions. The results are summarized:

(1)  A more detailed CFB emission inventory was generated in NCP. The daily CFB

emissions were estimated depending on CFB activities captured by MODIS.

Plenty of pollutants emitted from SNCP on Oct. 6$^{th}$ and 7$^{th}$, producing plenty of

PM$_{2.5}$ pollution, but few in NNCP during the entire haze period.

(2)  The WRF-CHEM model reproduced the pollution episode with a good agreement with observations. The correlation coefficients (R) of simulated and measured PM2.5 concentration are 0.88 in both NNCP and SNCP, and the related NMB are -12% in NNCP and -7% in SNCP. The simulated winds and

PBLH are also in good agreement with observations in both NNCP and SNCP.

(3)  The WRF-CHEM model was used to investigate the impacts of CFB

contribution and its evaluation on PM$_{2.5}$ concentration. The SNCP region is mainly influenced by the local CFB emissions, causing a maximum of 34%

PM$_{2.5}$ increase. Whereas the NNCP region is mainly affected by the long-range transport of pollution plume emitted from CFB in SNCP, causing a maximum of

32% PM$_{2.5}$ increase in NNCP.

(4)  The research domain includes two regions of interests. One is the NNCP, including two mega cities (Beijing and Tianjin), where few CFB occurred.

Another is the SNCP, where substantial CFB occurred. This study shows that there are substantially long-transport of CFB plume from SNCP to NNCP. More importantly, the effect of CFB remains in longer time in NNCP than in SNCP

along the foothill areas of the Taihang Mountains, causing significant enhancement in Beijing in both time and magnitude.

(5) Another major finding is that the mountains, surrounding the NCP in the north and west, play significant roles in enhancing the $PM_{2.5}$ pollution in NNCP through the blocking effect. Mountains block and redirect the airflows, causing the pollution accumulation along the foothill of mountains. The Taihang Mountains had greater impacts on $PM_{2.5}$ concentration than the Yanshan Mountains.

On account of various factors, such as pollutant long-range transport and pollutant accumulation caused by mountain effects, the prohibition of CFB should be strict not just in or around Beijing, but also on the ulterior crop growth areas of SNCP. Other $PM_{2.5}$ emissions in the SNCP should be significantly limited in order to reduce the occurrences of heavy haze events in NNCP region, including the Beijing City.

**Acknowledgement**

The PBL height and wind field data was obtained from the European Centre for Medium-Range Weather Forecasts (ECMWF) website (http://www.ecmwf.int/products/data/). This work is supported by the National Natural Science Foundation of China (NSFC) under Grant Nos. 41275186 and 41430424, and the Open Fund of the State Key Laboratory of Loess and Quaternary Geology (SKLLQG1413). The National Center for Atmospheric Research is sponsored by the National Science Foundation.

**Figure Captions**

Figure 1 The study area, sampling sites and crop fires. (a) The research domain and related provinces in China. (b) Topographical conditions of North China Plain. (c) Location of sampling sites and crop field burning captured by MODIS during the haze episodes. Green crosses indicate the measurement sites, and the CFB are shown by the pink dots.

Figure 2 The (a) yearly and (b) monthly crop field burning observed by MODIS in the research domain during the year of 2008 to 2014.

Figure 3 Crop field burning captured by MODIS with the background of MODIS real-time true color map (Left) and related CO emission (Right) on Oct. 6[th] and 7[th].

Figure 4 Regional averaged temporal variations in simulated (in red) and observed (in blue) results of species concentrations of (a) $PM_{2.5}$ (b) $O_3$ and (c) $NO_2$ over the regions of NNCP and SNCP.

Figure 5 Regional averaged temporal variations in simulated (in red) and observed (in blue) results of meteorological parameters of (a) wind speed (b) wind direction and (c) PBLH over the regions of NNCP and SNCP.

Figure 6 Backward trajectories of NNCP (Beijing, Tianjin and Baoding) during S1 (LST, 12:00 6th - 00:00 8th) in different height of 100m, 500m and 1000m.

Figure 7 Hourly $PM_{2.5}$ concentration of observations (obs) and simulations (sim-total and sim-anthro) in (a) NNCP and (b) SNCP. Sim-total represents the simulations considering total emissions (anthropologic and crop field burning), whereas sim-anthro is the simulations only considering anthropologic emissions.

Figure 8 CFB contribution to $PM_{2.5}$ concentration ($CPM_{2.5}$) (a) in SNCP, (b) in NNCP and (c) their comparison. The key point-in-local-times of T1 (23:00 6[th]), T2 (05:00 7[th]), T3 (20:00 7[th]) and T4 (19:00 8[th]) are signed with blue arrow.

Figure 9 The distributions of $TPM_{2.5}$ and $CPM_{2.5}$ of the key point-in-local-times of T1,

T2, T3 and T4, which represent different pollution phase of emission from crop field burning to $PM_{2.5}$. Left panels also show the pattern comparisons of simulated vs. observed near-surface $PM_{2.5}$ concentrations ($TPM_{2.5}$), with

$PM_{2.5}$ observations of colored circles. Black arrows denote simulated surface winds.

Figure 10 The elevation contours and the pattern comparisons of simulated vs.

observed near-surface $PM_{2.5}$ concentrations from 12:00 $7^{th}$ to 00:00 $10^{th}$.

Colored circles: $PM_{2.5}$ observations of foothill sites; Colored squares: $PM_{2.5}$

observations of non-foothill sites; Black arrows: simulated surface winds.

The 200-meter contour was highlighted with bold black line.

Figure 11 The averaged spatial distribution of $PM_{2.5}$ concentration and horizontal winds during 12:00 $7^{th}$ to 00:00 $10^{th}$. (a) Simulated $PM_{2.5}$ loading with erase behavior $RPM_{2.5}$, involving R-TY, R-T, and R-Y. (b) The related impacts of mountains to $PM_{2.5}$ ($IPM_{2.5}$), which represents the net effect of related mountains. The bold black lines were used to stress enclosing scope of each erased behavior.

Table 1. The average PM2.5 concentration, wind direction and wind speed of the observations from 12:00 6th to 00:00 12th. The sampling sites located at the foot of mountains were emphasized with bold style.

[revised manuscript text omitted]

Figure 1

[Figure]

Figure 2

[Figure]

Figure 3

[Figure]

Figure 4

[Figure]

Figure 5

[Figure]

Figure 6

[Figure]

Figure 7

[Figure]

Figure 8

[Figure]

Figure 9

[Figure]

Figure 10

[Figure]

Figure 11